# Wave-current interaction during Hudhud cyclone in the Bay of Bengal

Samiksha Volvaiker[1], Ponnumony Vethamony[1], Charls Antony[1], Prasad Bhaskaran[2], Balakrishnan Nair[3]

[1]Physical Oceanography Division, CSIR-National Institute of Oceanography, Dona Paula, 403004, Goa - India
[2]Department of Ocean Engineering and Naval Architecture, Indian Institute of Technology Kharagpur, Kharagpur - 721 302, India
[3]Indian National Centre for Ocean Information Services, Hyderabad - 500 090, India

*Correspondence to*: Samiksha S. V. (vsamiksha@nio.org)

**Abstract.** The present work describes the interaction between waves and currents utilizing a coupled ADCIRC+SWAN model for the very severe cyclonic storm 'Hudhud' which made landfall at Visakhapatnam on the east coast of India in October 2014. Model computed wave and surge heights were validated with measurements near the landfall point. The Holland model reproduced the maximum wind speed of $\approx$ 54m/s with the minimum pressure of 950hPa. The modelled maximum surge of 1.2 m matches with the maximum surge of 1.4 m measured off Visakhapatnam. The two-way coupling with SWAN showed that waves contributed $\approx$ 0.25m to the total water level during the Hudhud event. At the landfall point near Visakhapatnam, the East India Coastal Current speed increased from 0.5 to 1.8 m/s for a short duration ($\approx$6h) with net flow towards south, and thereafter reversed towards north. An increase of $\approx$0.2m in $H_s$ was observed with the inclusion of model currents. It was also observed that when waves travelled normal to the coast after crossing the shelf area, with current towards southwest, wave heights were reduced due to wave-current interaction; however, an increase in wave height was observed on the left side of the track, when waves and currents opposed each other.

## 1 Introduction

In coastal and shelf regions, winds and waves interact with the prevailing current system and several mutual non-linear interactions occur. Studies (Kudryavtsev et al., 1999; Davies and Lawrence, 1995; McWilliams et al., 2004) show that waves contribute to local currents, water level and mixing. Wind and wave induced currents can reinforce or interfere with tidal currents, depending on the phase of the tide. The impact of surface waves on currents or currents on waves is an important aspect in coastal hydrodynamics. Several studies have been carried out relating to individual processes, but not on interactions between them. Therefore, we need to take into account different processes that impact a specific process.

In the last few decades, there have been several efforts to develop theories and models on wave-current interactions (Ardhuin et al., 2008; Mellor, 2008; Warner et al., 2008; Uchiyama et al., 2010; Bennis et al., 2011). Holthuijsen and

Tolman (1991) and Komen et al. (1994) studied interaction between current and wave fields in the regions of the Gulf Stream, the Kuroshio and the Agulhas currents. The refraction theory of waves on current has advanced well, and this concept has been already introduced into the wave-action conservation equation. Linear wave theory on vertically sheared weak current is also discussed using both perturbation and numerical methods (Kirby and Chen, 1989; Dong, 2012). When waves propagate through strong currents, their characteristics change with refraction, bottom friction and blocking

(Kudryavtsev et al., 1999; Ris et al., 1999). Also, the mean flow will be affected by the addition of momentum and mass fluxes. With variation in water level, the depth felt by the waves also changes in the coastal region, thereby modifying the shallow water effects on the waves (Pleskachevsky et al., 2009). Liu et al. (2016) investigated the non-linear wave-current interaction in water of finite depth analytically using the homotopy analysis method (HAM) with solutions that are suitable for steep waves and strong currents expected during cyclonic conditions. The results were verified with flume experiments,

and the analytical solution was in good agreement with experimental results. Various parameters such as influence of water depth, wave steepness and current velocity on co-existing wave-current field were also reported in the above study.

Some of the wave processes that impact the coastal environment are as follows: (i) wave set-up during cyclones, which contributes significantly to storm surge and inundation; for example, when waves were included in the model, Beardsley et al., 2013 found that more areas were influenced by flooding in the Massachusetts Bay, (ii) wave-current

interaction increases the bottom friction, and thereby increasing the bottom stress. For example, Xie et al. (2001, 2003) introduced wave-induced surface and bottom stresses in the dynamic coupling between waves and currents, (iii) Carniel et al. (2009) and Zhang et al. (2011) included mixing due to wave breaking in their respective models and found improvements in the accuracy of surface drifter tracks in the Adriatic Sea and surface boundary layer thickness in the Yellow Sea, and (iv) Mellor (2003) and Xia et al. (2004) incorporated radiation stress in the coupling between wave, ocean

circulation and storm surge modelling.

Several numerical coupling experiments linking waves, currents and storm surges have been conducted in coastal areas in the past. For example, Tolman (1991) demonstrated the effect of water level and storm surges on wind waves for storms generated in the North Sea, and indicated that storm surges are essential factors to be considered for assessing the wave-current interactions. Mastenbroek et al. (1993) and Zhang and Li (1996) modelled the impact of waves on storm surges

and showed that wind stress with wave-dependant parameterization amplified the storm surge by 10–20%. Moon (2005) developed a wave-tide-circulation coupled system by including the influence of wave-current interaction, wave breaking and depth changes due to water level and found that the wave-dependent stress is strongly dependent on wave age and relative position from the storm centre. However, it may be noted that storm surge, tides or oceanic currents will have a significant effect on wave field only if their strengths are sufficient to interact.

Presently, in storm surge modelling, circulation and wave models are coupled in the same mesh, so that mesh resolution is fit to capture both circulation and wave physics. ADCIRC+SWAN (ADvanced CIRCulation + Simulating WAves Nearshore) is a coupled model that works on an unstructured mesh, and allows for interaction between storm

surges, waves and currents. This modelling system has been applied to hindcast hurricanes such as Katrina, Rita, Gustav and Ike (Westerink et al., 2008; Dietrich et al., 2011a, 2011b, 2012; Hope et al., 2013; Longley, 2013; Sebastian et al., 2014).

Several studies (Rao et al., 1982; Murty et al., 1986; Dube et al., 1997, 2000; Rao et al., 2013) reported storm surge along the east coast of India. Rao et al. (2012) simulated surge and inundation using ADCIRC for the following cyclones: Kavali (1989), Andhra (1996) and Cuddalore (2000). Three super cyclones, viz, 1999 Odisha cyclone, 2013 Phailin and 2014 Hudhud created significant impact along the east coast of India. Phailin cyclone generated waves with significant wave heights of the order of 7m (Balakrishnan et al., 2014). Hudhud was the first cyclone which effected urban areas and it is the

second severe cyclone which crossed the Visakhapatnam coast (Amarendra et al., 2015). Also, the beach erosion was very severe on the Ramkrishna beach, with a net sand volume of about 1457 cu.m lost over a stretch of 14 km (Hani et al., 2015). From the literature review, it is evident that most of the storm surge studies carried out  for the Indian coast used standalone models (Rao et al., 2012; Bhaskaran et al., 2014; Gayathri et al., 2015; Gayathri et al., 2016, Dhana Lakshmi et al., 2017). A comprehensive review on the coastal inundation research and an overview of the processes for the Indian coast was o

reported by Gayathri et al. (2017). One can find very few studies reported using a coupled model (ADCIRC with SWAN) for the Indian seas (Bhaskaran et al., 2013; Murty et al., 2014, 2016; Poulose et al., 2017) for extreme weather events. These studies examined the performance of coupled models and role of improved wind forcing on waves and hydrodynamic conditions. The coupled model (ADCIRC+SWAN) has demonstrated its efficacy in predicting storm surge and water level elevation as compared to the standalone ADCIRC model. For example,  the difference in residual water level at Paradeep

obtained by  standalone and coupled models at Paradeep in Odisha coast during 2013 Phailin cyclone were about 0.3m, and the coupled model performed relatively better than the standalone model (Murty et al., 2014). For the 2011 Thane cyclone also good performance of coupled parallel ADCIRC-SWAN model was reported by Bhaskaran et al. (2013). The model values of waves and currents obtained during Thane cyclone validated against HF Radar observations, satellite data  of ENVISAT, JASON-1, JASON-2 and wave rider buoy observations very clearly show that coupled model performed

reasonably well. During extreme weather events like cyclones, the interaction between waves and currents is a highly non-linear process, and the transfer and exchange of energy between them is a very complex process. Along the nearshore region, the non-linear interaction process is highly complex and to a larger extent, it is controlled by the local water depth and coastal geomorphological features. There can be instances, wherein the computed results using a coupled model may be under-estimated, when the influence of currents is considered. However, in this case the role of bottom characteristics and

water level needs a separate detailed study.

        The present study is a comprehensive exercise that aims at studying the following interactions during the Hudhud event: (i) impact of wave-current interaction on water level, (ii) impact of wave-current interaction on waves and (iii) impact of wave-current interaction on currents. This involves simulation of winds, tides, storm surges, currents and waves in the study domain during this extreme weather event using the coupled ADCIRC and SWAN models. Only  wave and water level

measured data were available for the verification of model results. Unfortunately, no measured current data was available for verification of the model-computed currents.

## 2. Data and methodology

### 2.1 Modelling system

ADCIRC and SWAN models were run in standalone and coupled modes on the same computational grid system. The cyclonic wind data were derived from the Holland formulation (Holland, 1980) using the best track estimate of Hudhud obtained from the JTWC (Joint Typhoon Warning Center) database. The hydrodynamic depth-averaged model ADCIRC applies the continuous Galerkin finite-element method to solve shallow water equations for water levels and vertically integrated momentum equations for velocity (Kolar et al., 1994; Atkinson et al., 2004; Luettich and Westerink,
2004; Dawson et al., 2006; Westerink et al., 2008; Kubatko et al., 2009; Tanaka et al., 2011). The model utilizes an unstructured mesh, and allows for refinement in areas where the solution gradients are the highest. It has an option for wetting and drying that activates and deactivates the entire grid elements during inundation and recession.

SWAN (Simulating WAves Nearshore) is a third-generation wave model developed at the Delft University of Technology, Netherlands. It computes random, short-crested wind-generated waves in coastal regions and inland waters
(Booij et al., 1999). The current version of SWAN is 40.85 (Zijlema, 2010). The model is based on the wave action balance equation, with various source and sink mechanisms, that governs the redistribution of energy balance in the wave system. SWAN can be used on any scale relevant for wind generated surface gravity waves. However, the SWAN model is specifically designed for coastal applications that should actually not require such flexibility in scale. The input parameters provided to SWAN includes bathymetry, current, water level, bottom friction and wind. The wave action balance equation is
expressed in the following form:

$$\frac{\partial N}{\partial t} + \frac{\partial C_{g,x} N}{\partial x} + \frac{\partial C_{g,y} N}{\partial y} + \frac{\partial C_{g,\sigma} N}{\partial \sigma} + \frac{\partial C_{g,\theta} N}{\partial \theta} = \frac{S}{\sigma}$$

where, N is the wave action density, $\sigma$ is the relative frequency, $\theta$ is the wave direction, Cg is the propagation speed in $(x,y,\sigma,\theta)$ space and S is the total of source/sink terms expressed as the wave energy density. In SWAN model, the source terms are expressed in the following form:

$$S = S_{in} + S_{ds,w} + S_{ds,b} + S_{nl4} + S_{nl3}$$

The terms in the R.H.S of the equation represent wind input, white-capping, bottom friction, quadruplet wave-wave
interactions and triad wave-wave interactions, respectively. The terms like bottom friction and triad wave-wave interaction can be neglected in deep water calculations. The model coupling is based on the work of Bunya et al. (2010) and Dietrich et

al. (2011) conducted for the Gulf of Mexico region. The SWAN model employs an implicit sweeping method to update the wave field at each computational vertex, which allows SWAN to apply longer time steps than ADCIRC. Thus, the SWAN time step usually defines the coupling interval between SWAN and ADCIRC models (Dietrich, 2010; Dietrich et al., 2011a,b). The wind field during Hudhud cyclone was generated using the Holland parametric model, which is specifically meant for simulating winds during cyclones.

The tide data were taken from the Permanent Service for Mean Sea Level (PSMSL) (www.psmsl.org). Wave data was obtained from the directional wave rider buoy deployed off Visakhapatnam (17.63°N; 83.26°E) at 15 m water depth. The measurement range is -20 m to 20 m, with an accuracy of 3%. The in situ data was recorded continuously at 1.28 Hz, and the recording interval for every 30 min was processed as one record. At every 200 s, a total number of 256 heave samples were collected and a Fast Fourier Transform (FFT) was applied to obtain a spectrum in the frequency range 0 to 0.58 Hz having a resolution of 0.005 Hz. Eight consecutive spectra covering 1600 s were averaged and used to compute the half-hourly wave spectrum. Significant wave height ($H_{m0}$) or $4\sqrt{m_0}$ was obtained from the wave spectrum. The $n^{th}$ order spectral moment ($m_n$) is given by: $m_n = \int_0^\infty f^n S(f)df$, where $S(f)$ is the spectral energy density at frequency $f$. The period corresponding to the maximum spectral energy (i.e., spectral peak period ($T_p$) was estimated from the wave spectrum. The wave direction ($D_p$) and directional width corresponding to the spectral peak were estimated based on the circular moments (Kuik et al.,1988).

## 2.2 Model domain and set-up

The model domain, chosen for the generation of winds, waves, currents and storm surges, covers the entire Bay of Bengal from 80-98°E and 6-21°N (Fig. 1a). The modified Etopo2 datasets by Sindhu et al. (2007) were used to generate the bathymetry grid. The data include improved shelf bathymetry for the Indian Ocean derived from sounding depths less than 200 m from the NHO (Naval Hydrographic Office, India) charts. The triangulated irregular mesh was prepared using SMS (Surface water Modeling System, http://www.aquaveo.com/) package for the selected domain (Fig. 1b). The unstructured mesh resolves sharp gradients in bathymetry, particularly in nearshore regions (Dietrich et al., 2011b), and it minimizes the computational cost relative to a structured mesh. For better results, tides and surges are resolved using a coarse grid in deep water, and higher resolution in the nearshore (Blain et al., 1994; Luettich and Westerink, 1995). Accordingly, in the present study, the mesh was generated with 82,253 elements and 41,795 nodes (Fig. 1b). A zoomed-in view of the landfall region with fine resolution of the mesh is shown in Fig. 1c. The mesh resolution varies from 1km in the nearshore region to a maximum of 80km in the deep water. The model has been run in a two-dimensional depth-averaged mode. The specifications of the model set-up are: (i) spherical coordinate system for the domain, (ii) cyclone duration (6.75 days), (iii)

constant bottom friction (0.0025), (iv) minimum depth of 0.5 m for wet and dry elements and (v) horizontal eddy viscosity coefficient of 2 m$^2$/s.

The dynamic Holland wind field model (Holland, 1980) calculates the wind field, sea-level pressure distribution and gradient wind within the tropical cyclone. The wind stress was specified to ADCIRC model using the relation proposed by Garrett (1977). Fig. 2 shows the relative position of cyclone eye and associated wind field of the Hudhud cyclone computed from the wind model at different intervals as the cyclone approached the coast, before making the landfall at Visakhapatnam coast. Holland model reproduced the maximum wind speed of ≈186 km/h with a minimum central pressure drop of 950 hPa when it transformed into a Very Severe Cyclonic storm.

## 2.3 Model setup for water level, current and wave generation

ADCIRC was tightly coupled to the unstructured wave model SWAN (Zijlema, 2010). The ADCIRC model was cold started with 13 tidal harmonic constituents (K1, N2, O1, P1, S2, K2, L2, M2, 2N2, MU2, NU2, Q1 and T2) taken from the LeProvost tidal database, and specified along the open boundary to reproduce tidal response in the Bay of Bengal. In the present study, the unstructured version of SWAN (version 40.85) has been used which implements the four-direction Gauss-Seidel iteration technique with unconditional stability (Zijlema, 2010). SWAN was discretized into 31 frequency bins ranging from 0.05 to 1.00 Hz on a logarithmic scale and 36 direction bins having an angular resolution of 10°. SWAN was setup with Cavaleri and Malanotte-Rizzoli (1981) wave growth physics; the shallow water triad non-linear interaction was computed using the lumped triad approximation of Eldeberky (1996). Earlier studies (Bhaskaran et al., 2014; Gayathri et al., 2015; Gayathri et al., 2016, Dhana Lakshmi et al., 2017; Bhaskaran et al., 2013; Murty et al., 2014, 2016; Poulose et al., 2017), carried out using the formulation of Komen et al. (1984) for cyclones which occurred in the Indian Ocean region, showed that SWAN with this scheme performed well for extreme weather events. Keeping this in view, in the present study, we have used the same formulation of Komen et al. (1984) to study the wave-current interaction during the Hudhud event. The model was initiated with modified white-capping dissipation (Komen et al., 1984); quadruplet non- linear wave-wave interaction was computed using Discrete Interaction Approximation (Hasselmann et al., 1985); depth induced breaking was computed using spectral version of the model with breaking index of γ = 0.73 (Battjes and Janssen, 1978); bottom friction was calculated based on JONSWAP physics (Hasselmann et al., 1973) with a friction coefficient, $C_b$ = 0. 05m$^2$s$^{-3}$. ADCIRC time step was specified as 10s, and SWAN as 600s. After every time step of SWAN, two-way coupling was carried out.

The model coupling is based on the work of Bunya et al. (2010) and Dietrich et al. (2011) in the Gulf of Mexico. SWAN employs an implicit sweeping method to update the wave details at each computational vertex, which allows SWAN to apply longer time steps than ADCIRC. Thus, the SWAN time step usually defines the coupling interval between SWAN and ADCIRC models (Dietrich, 2010; Dietrich et al., 2011a,b). SWAN computed radiation stress was passed on to ADCIRC to calculate wave set-up and nearshore currents. Similarly, water levels and currents computed by

ADCIRC were passed on to SWAN in the prescribed time step. SWAN accesses these inputs and wind speeds at each node and time, corresponding to the beginning and end of present interval. The radiation stress gradients used by ADCIRC were extrapolated forward in time, while the wind speeds, water levels and currents used by SWAN were averaged over each time step.

## 3. Results and Discussion

### 3.1. Cyclone track and wind generation

Hudhud cyclone is the second strongest tropical cyclone that crossed Visakhapatnam after 1985 (Amarendra et al., 2015) and caused extensive damage to the property. Hudhud crossed the Andaman Islands on 08 October 2014 at 0930h (IST). It moved west-northwest and intensified into a Very Severe Cyclonic Storm on 10 October 2014 (AN). It intensified further on 12 October and crossed the Visakhapatnam coast around 1300h (IST) with a maximum wind speed of 180 km/h (IMD Report, 2014). Figs. 1a and 2 show the track and passage of Hudhud. The maximum wind speed reproduced by the Holland model is ≈ 54 m/s (Fig. 2) with maximum pressure drop to 950 hPa.

### 3.2. Role of waves in surface elevation during Hudhud cyclone

Tidal phase plays a major role in affecting the surface elevation during cyclones. If a cyclone makes its landfall during high tide, the effective water level would be higher than during low tide. In this case, the landfall of Hudhud cyclone occurred during spring high tide. We have conducted three numerical experiments to assess the impact of waves, currents and tides on the total water surface elevation along the track during the passage of Hudhud cyclone. In the first experiment, the ADCIRC model was set-up with only the cyclonic winds and atmospheric pressure generated by the Holland Asymmetrical model (Fig. 2), and tides were switched-off. The model produced the maximum surge, which was due to cyclonic winds and pressure alone. In the second experiment, ADCIRC model was run with tides, cyclonic winds and atmospheric pressure, and the model provided the maximum water elevation generated by these contributing factors. The third experiment was a two-way coupling of ADCIRC and SWAN, that is, the model run was executed by combining winds, pressure fields, tides and wave forcing.

The resultant surface elevations from all these three numerical experiments were inter-compared and also validated with tide gauge data off Visakhapatnam. The tide data from the Permanent Service for Mean Sea Level (PSMSL) was adjusted to a Mean Sea Level (MSL) reference to match with ADCIRC generated surface elevation. Fig. 3 represents the spatial distribution of maximum water surface elevation (in the whole domain) produced by the cyclone from the above three experiments. The India Meteorological Department (IMD Report, 2014) reports a maximum water level of 1.6 m. However,

the tide gauge at Visakhapatnam recorded a maximum water level of 1.4 m. The simulation with winds, tides and waves predicted a water level of 1.2 m (Fig. 4), which matches reasonably well with the measured data as well as other model predictions (with a difference of 0.2 m during peak surge).

The two-way coupling with SWAN showed an increment of ≈0.15m in total water level near Visakhapatnam during the cyclone, which was contributed by waves to the total rise in water level. Wave set-up along the coast was caused as a result of waves generated by the storm that subsequently released momentum (radiation stress, Longuett-Higgins and Stewart, 1964) to the water column due to dissipation. Therefore, during storm events, water level rises not only by winds, but by waves also, though the magnitude is much less compared to the water level contributed by the winds and pressure.

Model results from both the runs were analysed to observe the change in storm surge height due to wave setup along the storm affected coastal regions, and the maximum change in the modelled surge height was ≈0.25m (≈ 20% of total surge height) between Visakhapatnam and Srikakulam (Fig. 3 b&c). Overall, the model prediction showed that during Hudhud cyclone wave induced setup had a significant impact on the total surge height, which provides an example of the importance of coupling wave and circulation model in predicting the total storm surge height accurately, especially during extreme

tropical cyclones.

### 3.3 Effect of wave-current interaction on currents

Currents in the study region generated during the Hudhud cyclone period were analyzed to study the impact of wave- current interaction on the local current system. The maximum current speed obtained from the three numerical experiments (model runs) are shown in Fig. 5. As current measurements were not available for the cyclone period, the model

produced velocity fields were analyzed and compared with earlier studies. In general, the East India Coastal Current (EICC) flows towards north along the east coast of India (ECI) during southwest monsoon. During northeast monsoon, the current reverses, and flows southward (Schott et al., 1994; Schott and McCreary, 2001; Shankar et al., 2002). On average, the maximum current speed along the ECI varies from 0.2 to 0.5 m/s (Mishra, 2010; Mishra, 2011; Panigrahi et al., 2010). Misra et al. (2013) observed through model simulations that tidal currents near the coast (water depth=20m)

increases gradually from south to north.

The present simulations predicted current speeds upto 0.5 m/s, and this range is consistent with the earlier studies. However, during the cyclone period, the two-way coupling (ADCIRC+SWAN) increased the current magnitude by 0.25 m/s (due to waves) along the cyclone track and near the landfall region. When the cyclone made its landfall near Visakhapatnam, the current speed increased from 0.5 to 1.8 m/s for a short duration (≈6h) with direction of flow towards south. After ≈6h of

landfall, current speed reduced to ≈0.1 m/s, with reversal of current (towards north) (Figs. 6 & 7). The current pattern shows semi-diurnal variation associated with tidal currents. The spatial distribution of current speed and direction during the

cyclone period driven by winds, tides and waves is given in Fig. 7, and it is very evident how the flow pattern changed with the passage of cyclone.

## 3.4 Effect of wave-current interaction on waves

Waves were modelled using SWAN alone and SWAN coupled with the ADCIRC to assess the impact of currents on the cyclone generated waves. Measured wave data were available only at one location, off Visakhapatnam (83.26°E, 17.63°N), which was on the track of Hudhud cyclone. Fig. 8 presents the comparison between the simulated and measured wave heights, wave periods and wave directions for the model runs of SWAN alone and coupled ADCIRC+SWAN. In the early stages of Hudhud, the wave heights were of the order of 3 -5m near the Andaman and Nicobar islands (Fig. 9). But,

when Hudhud intensified further while progressing towards ECI, it generated waves with heights of the order of 9-11 m, before making the landfall near Visakhapatnam on 12 October 2014 (1200h). Fig. 9 shows a swath of large waves (wave heights exceeding 10 m) propagating towards the coast with the passage of the storm. When the system was examined just before the landfall on 11 October 2014 at 2000 h (Fig. 9), it was found that the waves followed the pattern of cyclone winds. As waves experienced depth-limited breaking during its course onto the continental shelf, they propagated towards

the right side of the cyclone track. Near Visakhapatnam, the buoy recorded a peak wave height of 7.8 m (Fig. 8), whereas, the model peak value is 6.2m. Referring to Fig.8, we find that more or less the measured significant wave heights match with the modelled wave heights (with and without currents near the buoy location, off Visakhapatnam). When current was introduced, wave heights reduced approximately by 0.2m and mean wave periods reduced by 2s. It may be noted that during this time, the waves and currents were nearly in the same direction (Figs. 7 and 8d). Subsequently, when current speed

increased to 0.5 m/s (Fig. 6) during 1300h to 2000h (12th October 2014) with the wave and currents directions opposite to each other, we observe an increase in wave height of approximately 0.3m. Hence, there is an influence of currents on waves though it is marginal. The spatial distribution of maximum significant wave heights ($H_s$) simulated along the track of Hudhud cyclone using SWAN (no wave-current interaction) and coupled ADCIRC+ SWAN (with wave-current interaction) is given in Fig. 10 (a & b). Fig. 10(c) illustrates change in wave energy due to wave-current interaction. Table 1

highlights various statistical metrics with and without currents on the wave system at the buoy location for Hudhud. It is evident from Table 1 that inclusion of currents does not improve wave simulation for Hudhud cyclone. It is found that inclusion of currents deteriorated the wave simulation at the buoy location when waves and currents were nearly in the same direction, whereas, when waves and currents were in the opposite direction, the inclusion of currents enhanced the wave simulation. Overall, it is seen that the influence of currents on the wave system is marginal. This observation is also

supported by the recent study of Liu et al. (2016). They stated that an opposing current can lead to significant decrease in wave length and thereby tends to narrow both the crest and trough of the wave. This in turn causes an increased elevation in wave crest as the opposing current speed increases, whereas the wave trough elevation tends to remain constant throughout.

On the contrary, when waves and currents follow same direction, there is enhancement in wave length that tends to decrease the wave height elevation (Liu et al., 2016).

The spatial distribution of mean wave period ($T_m$) and peak wave period ($T_p$) simulated along the track of Hudhud cyclone using coupled ADCIRC+SWAN model (with wave-current interaction) is presented in Fig. 11 (a & b). Fig. 11a shows large mean wave periods (≈13s) in the nearshore region off Visakhapatnam during the cyclone (otherwise, during normal condition, wave periods will be of the order of 6s). Fig. 11b shows small pockets (at a few locations) of waves with large peak periods, of the order of 20s, moving towards the coast, south of Visakhapatnam. It was found that despite these large peak periods, the coupled wave-surge modelling system reproduced reasonably good wave-induced water level changes at these locations. Bender et al. (2012) reported similar large peak period scenarios, and reasoned that the ADCIRC model applies the SWAN radiation stress gradients based on individual spectral components only, and not the peak or mean parameters. This feature is also supported by the results of another coupled model, STWAVE, applied to the Louisiana Storm Surge (Atkinson et al, 2008), where isolated regions exhibited peak wave periods, greatly different from the surrounding values. Dietrich et al. (2013) presented a method that greatly removed the high peak period values with little degradation of model results. These isolated high peak wave periods point to the difficulty in simulating waves in inundating inland areas with shallow water depths and significant wind forcing.

Fig. 12a presents the maximum radiation stress gradient values calculated from SWAN, and passed on to the ADCIRC component of the coupled model. In the nearshore, the breaking waves exert stress on water column, causing changes in total water level and underlying currents. Fig. 12a shows the expected features for radiation stress gradient of 0.009 m$^2$s in the main wave breaking zone along the coastline when Hudhud made landfall between Visakhapatnam and Srikakulam.

We find from Fig. 10c that wave heights reduced by 0.5 m on the right side of the cyclone. Fig. 12b shows that waves travelled normal to the coast after crossing the shelf area, and currents flowed in the southwest direction (Fig. 7), and due to wave-current interaction wave heights have reduced. Subsequently, increase in wave height is noticed on the left side of the cyclone track when waves and currents opposed each other (waves propagated from southwest and currents flowed towards southwest direction, Fig. 7). In general, wave-current interaction is prominent, when currents are strong. The effect of currents on the wave field is examined by comparing the wave parameters collected off Visakhapatnam and the model results obtained from SWAN alone and ADCIRC+SWAN just before the landfall of the cyclone (Fig. 8). As discussed earlier, we observed an increase in current speed of≈1.3m/s just before the landfall (Fig. 6), and a an increase of ≈0.2m in the significant wave height ($H_s$).

## 4. Conclusions

A coupled ADCIRC+SWAN modelling system has been used to simulate the changes that occurred in the ocean surface dynamics during the passage of Very Severe Cyclonic Storm Hudhud that made landfall near Visakhapatnam, located on the east coast of India. At the time of peak intensity, the Holland parametric model reproduced maximum wind speed of ≈54 m/s with a minimum central pressure drop of 950 hPa. The landfall of Hudhud event occurred during the spring high tide, and the tide gauge observation off Visakhapatnam recorded a maximum surge of 1.4 m, that matched reasonably well with the modelled surge (1.2 m). The two-way coupling with SWAN showed an increment of ≈0.25 m (20%) in the total water level elevation, which was contributed by waves to the total rise in water level. During the time of landfall near Visakhapatnam, the current speed increased from 0.5 m/s to 1.8 m/s for a short duration (≈6 h) with the direction of flow towards south, and thereafter (≈ 6 h), the current speed reduced to ≈ 0.1 m/s with reversal in direction (towards north). The study signifies that an increase of ≈ 0.2 m in significant wave height ($H_s$) was noted when the effect of currents was included on the wave field. The inclusion of currents in the modelling system, thus has, influence on the wave field, especially on wave length (in the present case, a change of about 2 s in wave period) and wave height. Increase in wave height was observed on the left side of the cyclone track, when waves and currents opposed each other (waves were propagating from southwest and currents flowing towards southwest). As wave-current interaction is a complex problem, and the expected changes in wave parameters are very small, further refinement is required in the two-way coupling of ADCIRC+SWAN (with fine resolution bathymetry and improved cyclonic winds).

**Acknowledgements**

We thank Director, CSIR-NIO, Goa for his support and interest in this study. The first author acknowledges the Dept. of Sci & Tech, Govt. of India for supporting the research work through WOS-A(SR/WOS-A/ES-17/2012). The fieldwork data sharing is bounded with our institute data sharing policy. The ERA-Interim wind and wave data were freely downloaded from ECMWF (http://apps.ecmwf.int/datasets/). We are thankful to INCOIS, Hyderabad for providing the wave data. We acknowledge CSIR-NIO for providing high performance computing domain, HPC-Pravah for running the model. We are thankful to Dr. V.S.N Murty for giving input on impact of Hudhud on the coast. We are thankful to model developers for providing the source code for the model used in this study, ADCIRC+SWAN. We are also thankful to Chaitanya for assisting in preparation of the figures. The NIO contribution number is xxxx.

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

## Table 1. Statistical measures with (coupled)/without (standalone) currents on waves at the buoy location

| Statistical Metrics | Mean (m) | Bias (m) | RMSE (m) | Scatter Index | Correlation Coefficient |
|---|---|---|---|---|---|
| SWAN (standalone) | 1.89 | -0.08 | 0.53 | 0.28 | 0.95 |
| Coupled (ADCIRC+SWAN) | 1.89 | -0.04 | 0.48 | 0.25 | 0.95 |

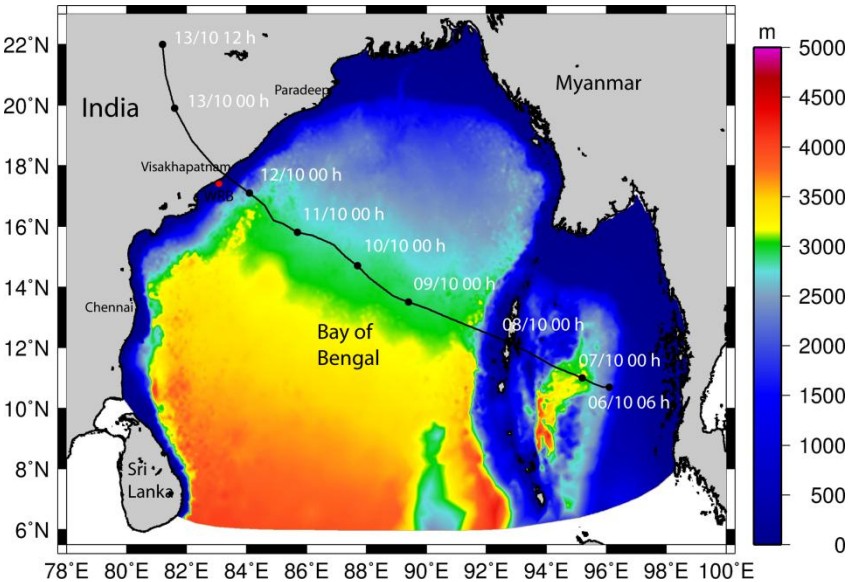

**Figure 1a**

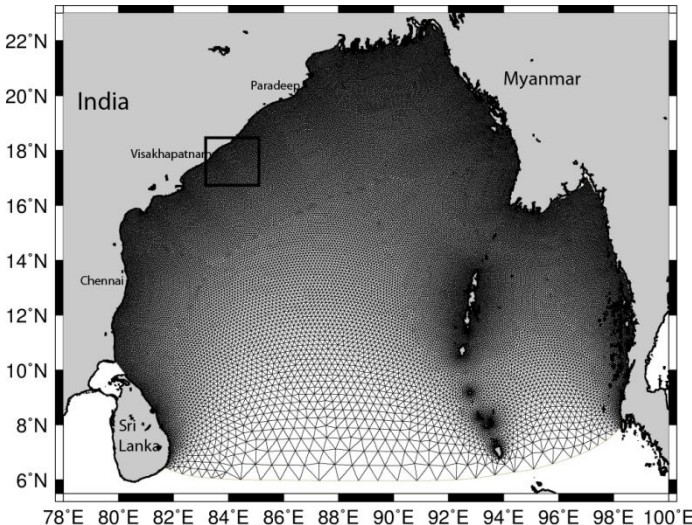

**Figure 1b**

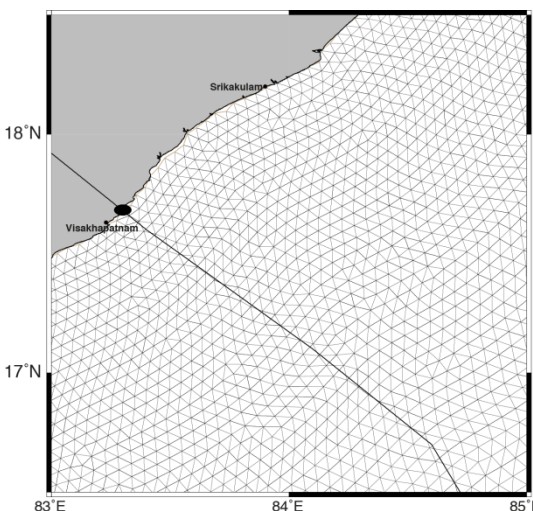

**Figure 1c**

**Fig. 1a. Bathymetry of the model domain chosen for wave-current interaction during Hudhud cyclone; cyclone track details are also shown; red dot represents wave rider buoy location. Fig. 1b. Fine resolution unstructured mesh generated for the domain to run the coupled ADCIRC+SWAN model; rectangular box represents the region where measured data are available for model validation (details of the box is shown in Fig. 1c). Fig. 1c. Fine-resolution mesh of the box shown in Fig. 1b; black circle is the**
**landfall point of the Hudhud cyclone; cyclone track is also shown.**

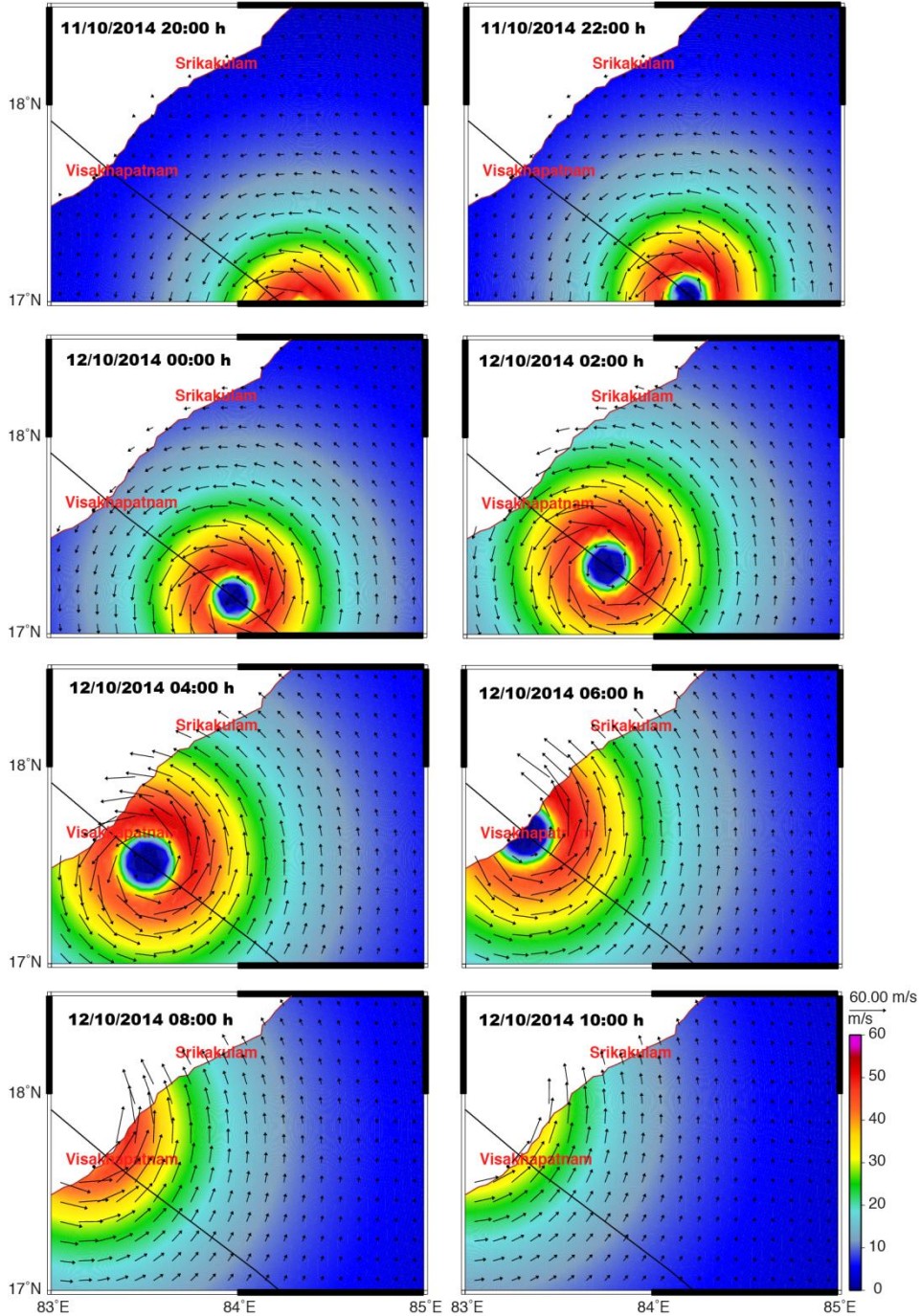

**Fig. 2. Typical winds (speed and direction) generated using Holland symmetrical model along the track of Hudhud cyclone (colour code represents wind speed in m/s; vectors represent wind direction).**

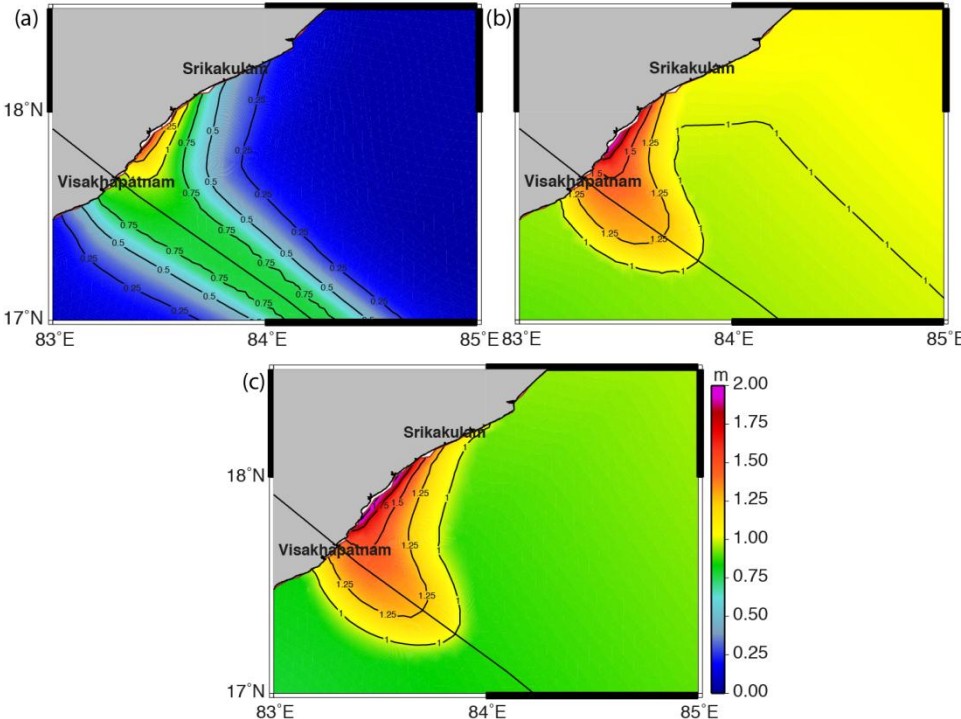

**Fig. 3. Spatial distribution of maximum surface elevation (m) due to (a) cyclonic winds, (b) cyclonic winds and tides and (c) cyclonic winds, tides and waves (colour code represents surface elevation in m).**

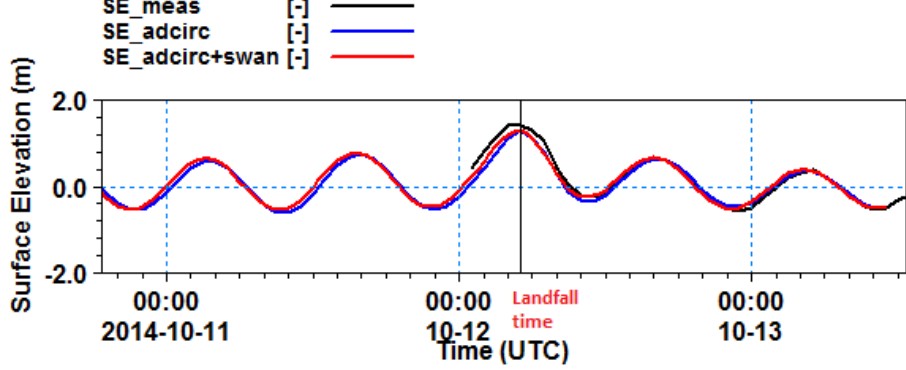


**Fig. 4. Time series of surface elevation (m) representing measured surface elevation (red line), SE from ADCIRC alone (blue line) and SE from ADCIRC+SWAN (black line) at Visakhapatnam coast (17.63°N; 83.26°E) during 10-13 October 2014.**

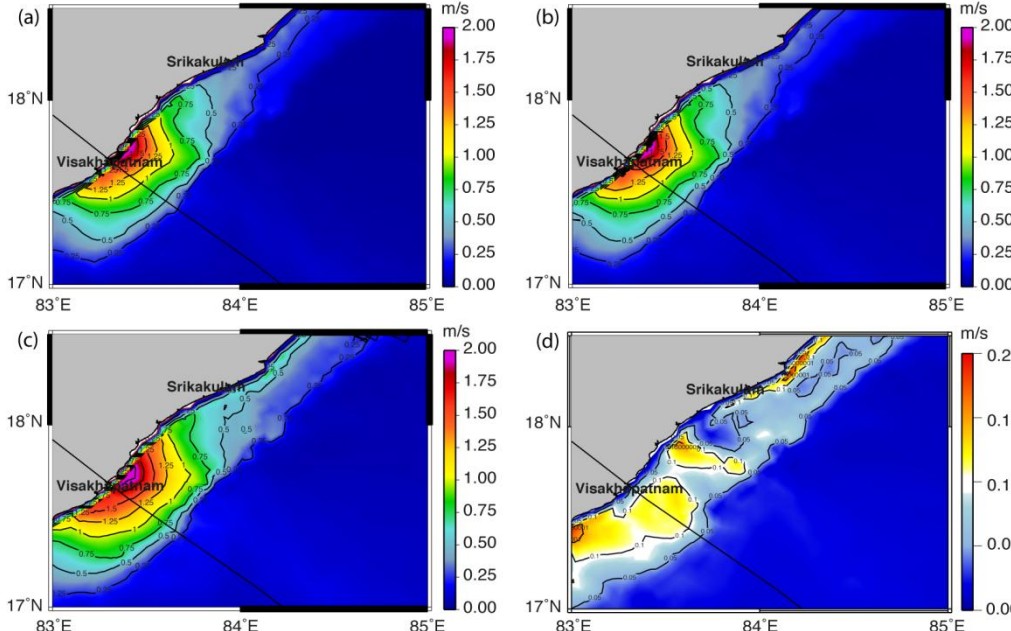

**Fig. 5. Spatial distribution of maximum surface currents (m/s) due to (a)  winds, (b) winds and tides and (c) winds, tides and waves, during cyclone, (d) difference in current speeds from (b) and (c), illustrating change in current speeds due to wave-current interaction (colour code represents current speeds in m/s).**

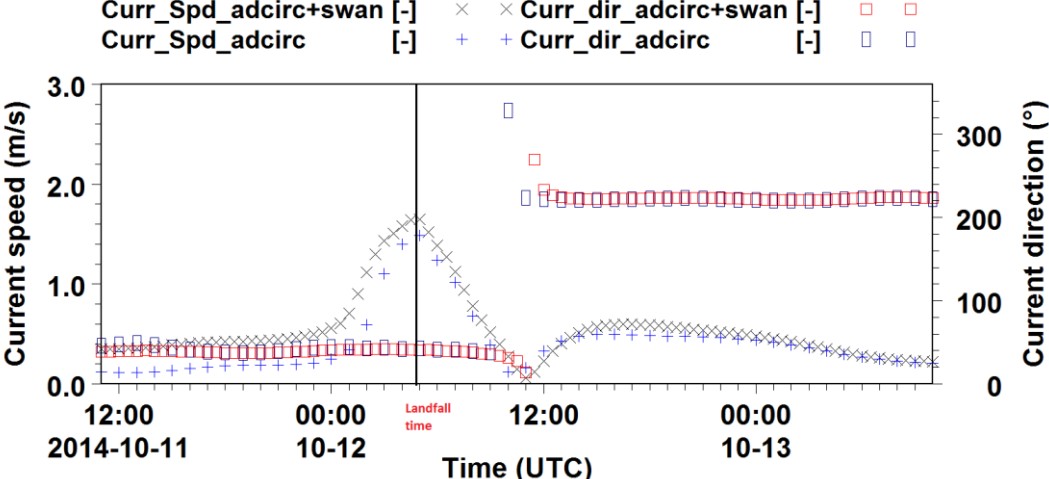

**Fig. 6. Time series of currents (m/s) representing current speeds and direction obtained from ADCIRC alone ('x' and blue rectangle) and coupled ADCIRC+SWAN ('+' and red rectangle) off Visakhapatnam coast (17.63°N; 83.26°E) during 10-13 October 2014.**

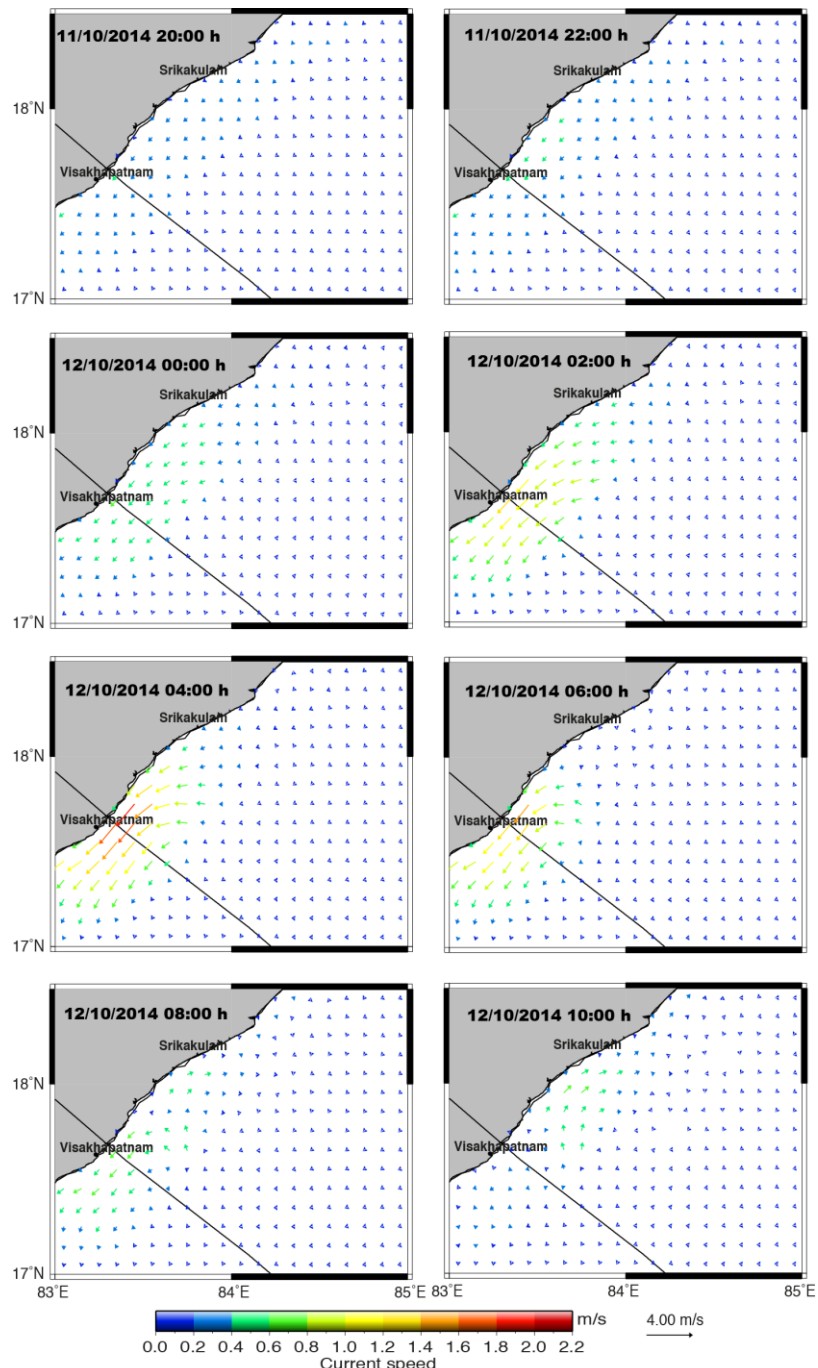


**Fig. 7. Current speed and direction simulated along the track of Hudhud cyclone using the coupled ADCIRC+SWAN model (colour code represents current speed in m/s; vectors represent current direction).**

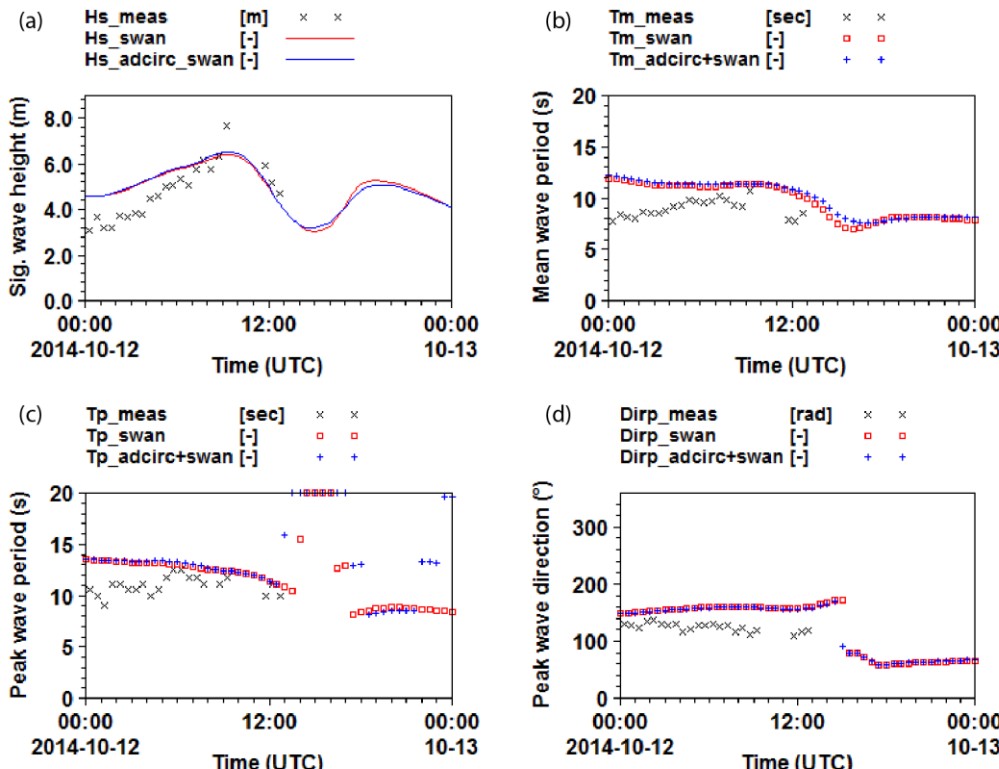

**Fig. 8. Comparison of measured (black) and modelled (a) significant wave heights (H<sub>s</sub>), (b) mean wave periods, (c) peak wave periods and (d) peak wave directions obtained from SWAN (red) and coupled ADCIRC+SWAN (blue) during Hudhud cyclone with measured data off Visakhapatnam (17.63°N; 83.26°E).**

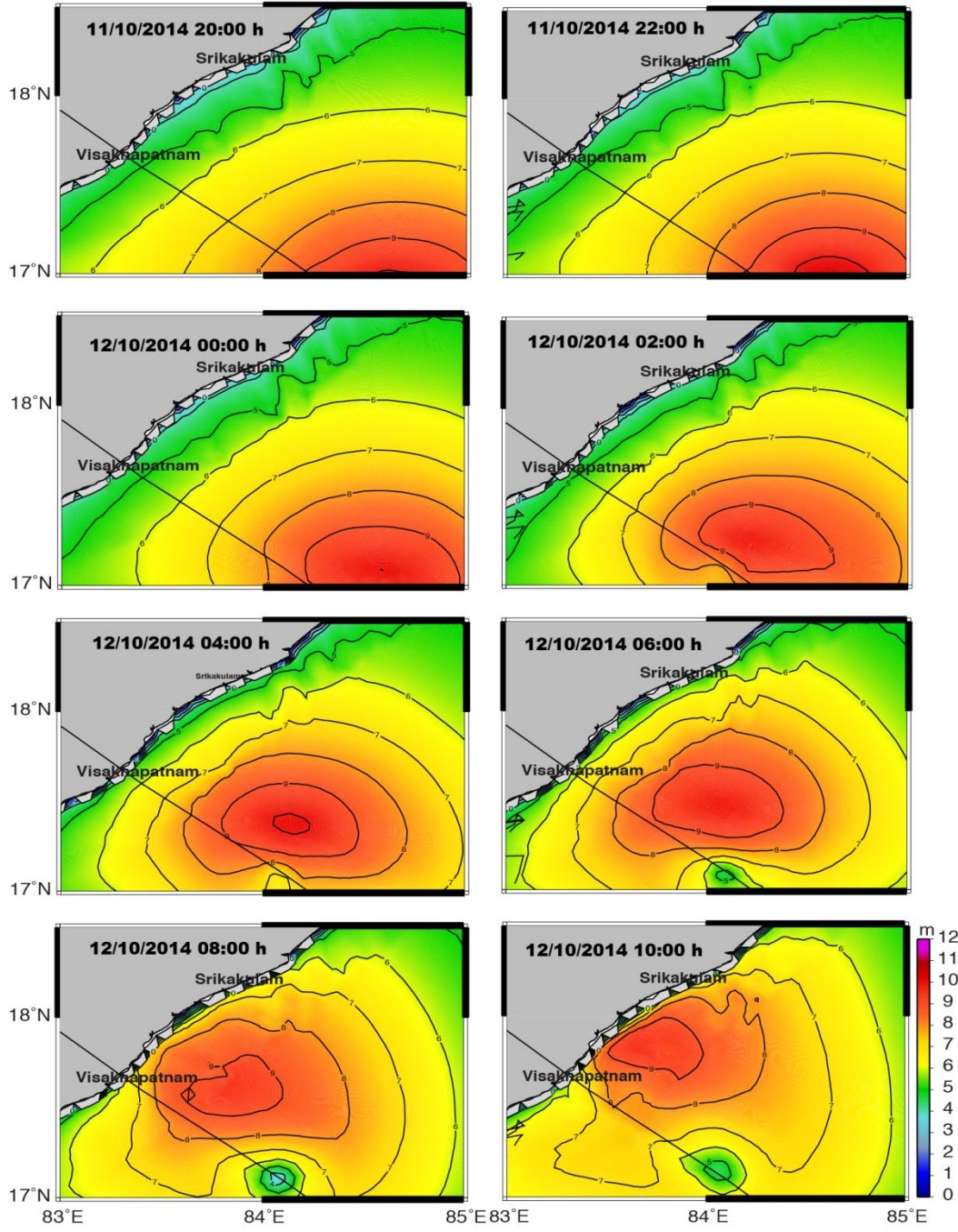


**Fig. 9. Significant wave heights (H$_s$) simulated along the track of Hudhud cyclone using the coupled ADCIRC+SWAN model (colour contours represent H$_s$ in m).**

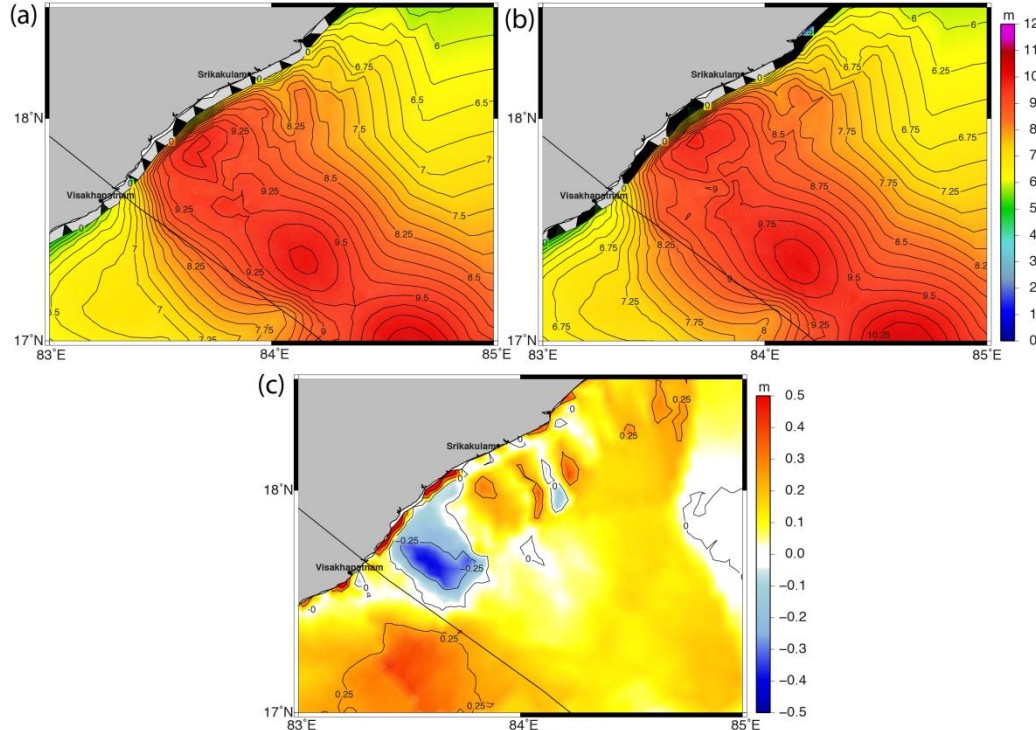

**Fig.10. Spatial distribution of maximum significant wave heights (H$_s$) simulated along the track of Hudhud cyclone using (a) SWAN model (no wave-current interaction), (b) coupled ADCIRC+SWAN model (with wave-current interaction); colour code and contours represent H$_s$; (c) change in H$_s$ from (a) and (b), illustrating change in wave energy due to wave-current interaction.**

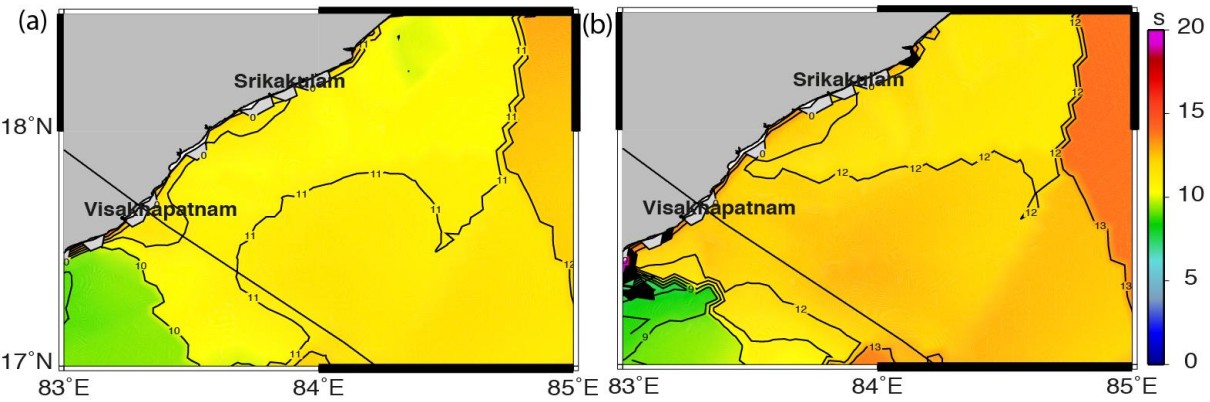

**Fig. 11. Spatial distribution of (a) mean wave period (T$_m$) and (b) peak wave period (T$_p$) simulated along the track of Hudhud**
**cyclone using coupled ADCIRC+SWAN model (with wave-current interaction).**

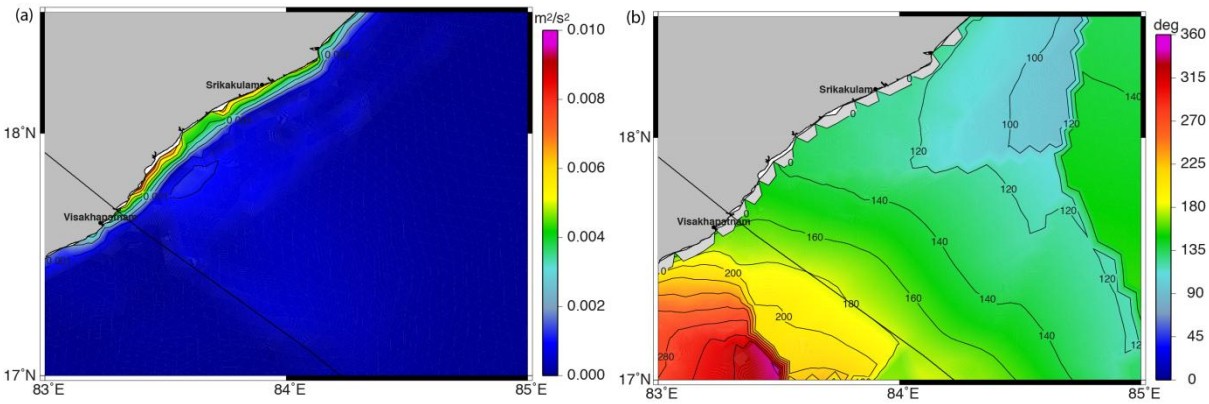

**Fig. 12. (a). Maximum radiation stress gradient values calculated from SWAN and (b) spatial distribution of mean wave direction (Dir) simulated along the track of Hudhud cyclone using the coupled ADCIRC+SWAN model (with wave-current interaction); colour code and contours represent wave direction.**