# Peer review of "Wave-current interaction during Hudhud cyclone in the Bay of Bengal"

_Natural Hazards and Earth System Sciences, 2017_

## Referee Comment (RC1) · Anonymous Referee #1 · 30 May 2017

The manuscript has a subject that is appropriate to the NHESS publication and should be of interest to readers of the journal. The paper is reasonably well written, logically organised, structured and illustrated. The authors present an interesting set of simulations to assess the mutual affects of waves and currents during the passage of the Very Severe Cyclonic Storm Hudhud. The principal results are: (1) Waves contributed 0.25 m to the total water level during the event, which agrees with measurements at Visakhapatnam; (2) Current speed increased from 0.5 to 1.8 m/s for a short time during the event; (3) the two-way coupling increased the current magnitude by 0.25 m/s along the track; (4) The use of wave-ocean coupling increases Hs in 2 m compared to wave model only; (5) waves decrease due to currents when they travel normal to the coast after crossing the shelf area (right side of track) and increased on the left side of track when currents oppose wave direction. Cyclonic systems currently pose

one of the most challenging and important meteo-oceanographic phenomena for the earth science community and the study is a valuable contribution. However I have fundamental objections that I believe should be addressed before final publication.

My main concern is related to the overall content and discussions. In the end of introduction, the authors state that "The present study primarily aims at quantifying the impact of wave-current interaction on waves during the Hudhud cyclone". The analysis however seems to be equally focused on the effects of waves on currents and water level. I presume they do so because the only data source available is of a Wave Rider buoy. No current data is available. However, there is no clear discussion on whether the inclusion of current improves wave simulation at the buoy location. The authors just mention the differences in model results show the plots of comparison of model and measurement and let the readers draw their own conclusion. The coupling system increase wave height (0.2 m) at the wave height peak moment. But wave height is decreased before this moment and model actually agrees better with data without the inclusion of currents. Wave period is also slightly better represented in the simulation without currents. If the main goal of the paper is "quantifying" the effect of wave-current interactions on waves, this must be discussed also in terms of improvement and/or deterioration of simulations compared to measurements. At least an attempt should be made. It is an interesting opportunity to address some limitations of these models and if currents are actually beneficial to wave modelling (and vice-versa).

Other specific concerns are: (1) The exact location of the wave rider buoy must be plotted, possibly on the map of figure 1a, so that the reader can know where the validation of the wave model was performed for.

(2) Very little detail is given about wave measurements. Although section 2 is entitled "Data and Methodology" it is basically about the modelling configuration. What is the sampling time of wave information, how is it obtained (spectral method, record length)? This information together with the aforementioned plot of the buoy location may be of interest to readers.

(3) Why do the authors decide for the set of physics of growth and dissipation from Cavaleri and Malanotte-Rizzoli (1981) and Komen et al. (1984)? These old parameterisations and especially the Komen et al. dissipation form are proven to not be adequate in non-standard conditions, as in opposing currents, for example (see Ardhuin et al JPO (2012) and Rapizo et al (JGR 2017)). The dissipation form in Westhuysen et al (2007) shows better performance than the Komen et al. term in adverse currents (Rapizo et al, 2017). The newly implemented in SWAN and recently released 'ST6' physics (Rogers et al, 2012) performs best in conditions of effective currents Rapizo et al (JGR 2017), which is the subject of investigation here. If the old physics are used instead, a justification must be given.

(4) I find it hard to analyse the differences in current speed shown in Fig. 5 (especially for figure (b)). Although it is interesting to see the pattern produced by the cyclone landfall, all figures show similar patterns. I suggest here to plot (b) and (c) as current speed differences (similar to Fig. 10 bottom panels).

Other minor points: Line 22: "Studies show that waves contribute to local currents, water level and mixing." By mentioning "Studies" I feel at least one reference is needed here.

Line 24-26: "Several studies have been carried out relating to individual processes, but not many on interaction between the processes. Therefore, we need to take into account different processes that impact a specific process." Very confusing, many repetitions of word "process". Rephrase.

Line 34: "effected" => "affected"

Line 37: "The wave processes that impact the coastal environment are:" There are many other wave-related processes that impact the coastal environment other than the ones listed (wave set-up, wave-current interactions and breaking-induced mixing). The first phrase should be rephrased to something like: "Some of the relevant wave processes that impact the coastal environment are as follows:"

[Figure]

Line 55: SWAN stands for Simulating Waves Nearshore, not "in Nearshore".

Line 93: Buoy coordinates are wrong. (same for legend in Fig. 4, 6 and 8)

---

## Referee Comment (RC2) · Anonymous Referee #2 · 6 Jun 2017

The paper at hand is interesting, and addresses an important topic, namely the influence of wave-current interactions on water set up, current magnitude, and water wave evolution in very strong conditions. The case studied here is the Hudhud cyclone. Obviously, the topic is relevant for publication in Natural Hazards and Earth System Sciences. Furthermore, the paper is clear and well structured. It is relatively well written, and pleasant to read. For these reasons, I believe it should be published in NHESS. However, I have some concerns, which, if addressed, could help improve the paper.

1) The introduction states that "The present study primarily aims at quantifying the impact of wave-current interaction on waves during the Hudhud cyclone". But the paper presents more results than this (effect of wave-current coupling in the modelling tech-

nique for predicting set-up, current, or waves). Later, the discussion clearly focusses on the modelling technique, and the influence of coupling wave and currents from both models. This issue is more technical, but also really interesting. Finally, the conclusion comes back on the topic suggested in introduction. I suggest the authors slightly modify introduction and conclusion to mention both type of results in introduction and conclusion.

2) The modelling procedure for SWAN could be detailed a little bit more. The conditions used here are really not usual conditions, and a commentary on how accurate the approximations are in hurricane conditions would be welcomed.

3) Except for the wave buoy data (I would appreciate to see the location of the buoy on a map, by the way), the paper suffers a lack of data for validation. Could the authors access some other data, such as surface velocity from satellite, or water elevation from PSMSL, for instance? It would help validating the numerical results.

4) In section 3.4, I had difficulties to understand if the "SWAN alone" simulations were referring to SWAN with absolutely no current, or SWAN with input current from AD-CIRC, but no coupling. This clarification is obviously important for interpreting the results.

Minor points:

- section 2.1, line 94. There is a misprint on the location of the wave rider buoy. Furthermore, I could not understand what the +20m -20m measurement range refers to. Wave Height? It seams huge, and is probably not true regardless to the waves frequency.

- Section 3.1: Do we have an estimation on how accurate Holland's numerical results were? Could the authors mention it with a sentence?

- For every figure, the captions are not detailed enough. Most of the time, it is unclear what symbol corresponds to what line. The date and time used for various maps are not mentioned.

---

## Author Comment (AC1) · 27 Jun 2017

RESPONSE SHEET - REVIEW COMMENTS
The manuscript has a subject that is appropriate to the NHESS publication and shouldbe of interest to readers of the journal. The paper is reasonably well written, logicallyorganised, structured and illustrated. The authors present an interesting set

of simulationsto assess the mutual affects of waves and currents during the passage of theVery Severe Cyclonic Storm Hudhud. The principal results are: (1) Waves contributed0.25 m to the total water level during the event, which agrees with measurements atVisakhapatnam; (2) Current speed increased from 0.5 to 1.8 m/s for a short time duringthe event; (3) the two-way coupling increased the current magnitude by 0.25 m/salong the track; (4) The use of wave-ocean coupling increases Hs in 2 m comparedto wave model only; (5) waves decrease due to currents when they travel normal tothe coast after crossing the shelf area (right side of track) and increased on the leftside of track when currents oppose wave direction. Cyclonic systems currently poseone of the most challenging and important meteo-oceanographic phenomena for theearth science community and the study is a valuable contribution. However I havefundamental objections that I believe should be addressed before final publication.

Response: The authors thank the reviewer for thoroughly reviewing our manuscript, appreciating the work and providing valuable suggestions to improve the manuscript. The fundamental objections raised by the reviewer have been addressed in the following paragraphs.

Comment – 2:

My main concern is related to the overall content and discussions. In the end of introduction,the authors state that "The present study primarily aims at quantifying theimpact of wave-current interaction on waves during the Hudhud cyclone". The analysishowever seems to be equally focused on the effects of waves on currents and waterlevel. I presume they do so because the only data source available is of a Wave Riderbuoy. No current data is available. However, there is no clear discussion on whetherthe inclusion of current improves wave simulation at the buoy location. The authorsjust mention the differences in model results show the plots of comparison of modeland measurement and let the readers draw their own conclusion. The coupling systemincrease wave height (0.2 m) at the wave height peak moment. But wave height isdecreased before this moment and model actually agrees better with data without

theinclusion of currents. Wave period is also slightly better represented in the simula-tionwithout currents. If the main goal of the paper is "quantifying" the effect of wave-currentinteractions on waves, this must be discussed also in terms of improvement and/or deteriorationof simulations compared to measurements. At least an attempt should bemade. It is an interesting opportunity to address some limitations of these models andif currents are actually beneficial to wave modelling (and vice-versa).

Response: The authors appreciate the reviewer comments and agree that this study focus not only on the quantification pertaining to the impact of wave-current interac-tion, but also on: (i) impact of wave-current interaction on water level, (ii) impact of wave-current interaction on waves and (iii) impact of wave-current interaction on cur-rents. Accordingly, the last paragraph in the Introduction section has been modified, and relevant references were added as follows:

From literature review, it is evident that most of the studies carried out with storm surge models for the Indian coast used standalone models (Rao et al., 2012; Bhaskaran et al., 2014; Gayathri et al., 2015; Gayathri et al., 2016, Dhana Lakshmi et al., 2017). A comprehensive review on the coastal inundation research and an overview of the processes for the Indian coast was also reported by Gayathri et al. (2017). One can find very few studies reported using a coupled model (ADCIRC with SWAN) for the Indian seas (Bhaskaran et al., 2013; Murty et al., 2014, 2016; Poulose et al., 2017) for extreme weather events. These studies examined the performance of coupled models and role of improved wind forcing on waves and hydrodynamic conditions. The present study is a comprehensive exercise that aims to study the following interaction during the Hudhud event: (i) impact of wave-current interaction on water level, (ii) impact of wave-current interaction on waves, and (iii) impact of wave-current interaction on cur-rents. This involves simulation of winds, tides, storm surges, currents and waves in the study domain during this extreme weather event using the coupled ADCIRC and SWAN models. Only the measured wave and water level data was available for the verification of model results (which happened to be very close to the cyclone track).

Both these data sets were utilized in this study. Unfortunately, no measured current data was available for verification of the model-computed currents. The coupled model (ADCIRC+SWAN) has demonstrated its efficacy in predicting storm surge and water level elevation as compared to the standalone ADCIRC model. For example, considering the 2013 Phailin cyclone event (Murty et al., 2014), the difference in residual water level between standalone and coupled versions at Paradeep in Odisha coast were about 0.3m, and the coupled model performed relatively better than standalone model. In addition, for the 2011 Thane cyclone, good performance of coupled parallel ADCIRC-SWAN model was reported by Bhaskaran et al. (2013). The overall performance of waves and currents during Thane event validated against HF Radar observations and with satellite tracks of ENVISAT, JASON-1, JASON-2 and wave rider buoy observations very clearly show that coupled model performed reasonably well. During extreme weather events like cyclones, the interaction between waves and currents is a highly non-linear process, and the transfer and exchange of energy between them is a very complex process. Along the nearshore regions, the non-linear interaction process is highly complex and to a larger extent, it is controlled by the local water depth and coastal geomorphological features. There can be instance wherein the computed results using a coupled model may be under-estimated considering the influence of currents. However, in this case the role of bottom characteristics and water level needs a separate detailed study. Also, including fine resolution bathymetry and cyclonic winds will further enhance the accuracy of the model.

References

Dhana Lakshmi, D., Murty, P.L.N., Bhaskaran, P.K., Sahoo, B., Srinivasa Kumar, T., Shenoi, S.S.C., Srikanth, A.S.: Performance of WRF-ARW winds on computed storm surge using hydrodynamic model for Phailin and Hudhud cyclones. Ocean Engineering, 131, 135-148, 2017.

Gayathri, R., Bhaskaran, P.K., Sen, D.: Numerical study on Storm Surge and associated Coastal Inundation for 2009 AILA Cyclone in the head Bay of Bengal. Int. Conf.

[Figure]

on Water Resources, Coastal and Ocean Engineering (ICWRCOE 2015), Aquatic Procedia, 4, 404-411, 2015.

Gayathri, R., Murty, P.L.N., Bhaskaran, P.K., Srinivasa Kumar, T.: A numerical study of hypothetical storm surge and coastal inundation for AILA cyclone in the Bay of Bengal. Environmental Fluid Mechanics, 16(2), 429-452, 2016.

Gayathri, R., Bhaskaran, P.K., Jose, F.: Coastal inundation research: an overview of the process. Current Science, 112(2), 267-278, 2017.

Murty, P.L.N., Bhaskaran, P.K., Gayathri, R., Sahoo, B., Srinivasa Kumar, T., SubbaReddy, B.: Numerical study of coastal hydrodynamics using a coupled model for Hudhud cyclone in the Bay of Bengal. Estuarine, Coastal and Shelf Science, 183, 13-27, 2016.

Poulose, J., Rao, A.D., Bhaskaran, P.K.: Role of continental shelf on nonlinear interaction of storm surges, tides and wind waves: An idealized study representing the west coast of India. Estuarine, Coastal and Shelf Science, http://dx.doi.org/10.1016/j.ecss.2017.06.007

Comment – 3:

The exact location of the wave rider buoy must be plotted,possibly on the map of figure 1a, so that the reader can know where the validationof the wave model was performed for.

Response: The authors appreciate the reviewer comments. Accordingly as suggested, the buoy location is marked in Figure 1a.

Comment – 4:

Very little detail is given about wave measurements. Although section 2 is entitled"Data and Methodology" it is basically about the modelling configuration. What is thesampling time of wave information, how is it obtained (spectral method, record length)?

[Figure]

This information together with the aforementioned plot of the buoy location may be of interest to readers.

Response: The authors appreciate the reviewer comments. The wave data used in this study was obtained from the National Institute of Ocean Technology, Chennai. As suggested by the reviewer, the details of wave measurements and data analysis are now added in the revised manuscript as follows:

Wave data was obtained from the directional wave rider buoy deployed off Visakhapatnam (17.63ïČřN; 83.26ïČřE) at 15 m water depth. The measurement range is -20 m to 20 m, with an accuracy of 3%. The in situ data was recorded continuously at 1.28 Hz and the recording interval for every 30 min was processed as one record. At every 200 seconds, a total number of 256 heave samples were collected and a Fast Fourier Transform (FFT) was applied to obtain a spectrum in the frequency range 0 to 0.58 Hz having a resolution of 0.005 Hz. Eight consecutive spectra covering 1600 seconds were averaged and used to compute the half-hourly wave spectrum. Significant wave height (H_m0) or $4\sqrt{(m\_0)}$ was obtained from the wave spectrum. The nth order spectral moment (mn) is given by: m_n=∫ _⊕∞âŰŠãĂŰfˆn S(f)dfãĂŮ, where S(f) is the spectral energy density at frequency f. The period corresponding to the maximum spectral energy (i.e., spectral peak period (T_p) is estimated from the wave spectrum. The wave direction (D_p) and directional width corresponding to the spectral peak is estimated based on the circular moments (Kuik et al.,1988).

Comment – 5:

Why do the authors decide for the set of physics of growth and dissipation fromCavaleri and Malanotte-Rizzoli (1981) and Komen et al. (1984)? These old parameterisationsand especially the Komen et al. dissipation form are proven to not be adequatein non-standard conditions, as in opposing currents, for example (see Ardhuin et al JPO(2012) and Rapizo et al (JGR 2017)). The dissipation form in Westhuysen et al (2007)shows better performance than the Komen et al. term in adverse currents (Rapizoet al, 2017).

The newly implemented in SWAN and recently released 'ST6' physics(Rogers et al, 2012) performs best in conditions of effective currents Rapizo et al (JGR2017), which is the subject of investigation here. If the old physics are used instead, ajustification must be given.

Response: The authors appreciate the reviewer comments. The authors have conducted this study in 2015 using the unstructured version of SWAN (version 40.85) implementing an analog to the four-direction Gauss-Seidel iteration technique with unconditional stability (Zijlema, 2010). However, Rapizo et al (2017) reported the good performance of SWAN in tidal current regime (ebb and flood flows) very recently (2017) only. It may kindly be noted that, the co-author of this work, Bhaskaran and his team has carried out a few studies (Bhaskaran et al., 2014; Gayathri et al., 2015; Gayathri et al., 2016, Dhana Lakshmi et al., 2017; Bhaskaran et al., 2013; Murty et al., 2014, 2016; Poulose et al., 2017) using the same formulation of Komen et al. (1984) for cyclones that occurred in the Indian Ocean region, and found that SWAN with this scheme performed well for extreme weather events also. Keeping this in view, in the present study the authors have gone ahead with using the same formulation of Komen et al to study the wave-current interaction during the Hudhud event. However, the authors appreciate the reviewer comments and shall use the scheme of Roger et al (2012) in SWAN and study the wave-current interaction in tidal as well in cyclonic conditions as a separate study in future.

Comment – 6:

I find it hard to analyse the differences in current speed shown in Fig. 5 (especiallyfor figure (b)). Although it is interesting to see the pattern produced by the cyclonelandfall, all figures show similar patterns. I suggest here to plot (b) and (c) as currentspeed differences (similar to Fig. 10 bottom panels).

Response: The authors appreciate the reviewer comments. As suggested one more Figure (5d) is added to show the difference in current speed similar to Fig. 10 in the

revised manuscript.

Comment – 7:

Other minor points: Line 22: "Studies show that waves contribute to local cur-
rents,water level and mixing." By mentioning "Studies" I feel at least one reference
is neededhere.

Response: The authors appreciate the reviewer comments. As suggested the following
three references are added to this statement in the revised manuscript:

Kudryavtsev et al., 1999; Davies and Lawrence, 1995; McWilliams et al., 2004. These
studies show that waves contribute to local currents, water level and mixing.

Comment – 8:

Line 24-26: "Several studies have been carried out relating to individual processes,but
not many on interaction between the processes. Therefore, we need to take intoac-
count different processes that impact a specific process." Very confusing, manyrepeti-
tions of word "process". Rephrase.

Response: The sentence is rephrased as follows in the revised manuscript: 'Several
studies have been carried out relating to individual processes, but not on the interac-
tions between them'.

Comment – 9:

Line 34: "effected" => "affected"

Response: The correction made accordingly in the revised manuscript.

Comment – 10:

Line 37: "The wave processes that impact the coastal environment are:" There are-
many other wave-related processes that impact the coastal environment other thanthe
ones listed (wave set-up, wave-current interactions and breaking-induced mixing).The

first phrase should be rephrased to something like: "Some of the relevant wavepro-cesses that impact the coastal environment are as follows:"

Response: The sentence is modified as follows in the revised manuscript: Some of the wave processes that impact the coastal environment are as follows: wave set-up, wave-current interactions and breaking-induced mixing.

Comment – 11:

Line 55: SWAN stands for Simulating Waves Nearshore, not "in Nearshore".

Response: The correction is made accordingly in the revised manuscript.

Comment – 12:

Line 93: Buoy coordinates are wrong. (same for legend in Fig. 4, 6 and 8)

Response: The authors appreciate the reviewer comments. As suggested, the corrections are made (17.63ïĆřN; 83.26ïĆřE) in the revised manuscript.

Please note that two figures are included in supplement file.

Please also note the supplement to this comment:
https://www.nat-hazards-earth-syst-sci-discuss.net/nhess-2017-11/nhess-2017-11-AC1-supplement.pdf

---

## Author Comment (AC2) · 27 Jun 2017

RESPONSE SHEET - REVIEW COMMENTS
The paper at hand is interesting, and addresses an important topic, namely the influenceof wave-current interactions on water set up, current magnitude, and water

waveevolution in very strong conditions. The case studied here is the Hudhud cyclone.Obviously, the topic is relevant for publication in Natural Hazards and Earth SystemSciences. Furthermore, the paper is clear and well structured. It is relatively well-written, and pleasant to read. For these reasons, I believe it should be published inNHESS. However, I have some concerns, which, if addressed, could help improve thepaper.

Response: The authors thank the reviewer for thoroughly reviewing our manuscript, appreciating the work and providing the positive recommendations. Based on the valuable suggestions, the authors have improved the manuscript. The response to comments and concerns are addressed below.

Comment – 2:

The introduction states that "The present study primarily aims at quantifying the impactof wave-current interaction on waves during the Hudhud cyclone". But the paperpresents more results than this (effect of wave-current coupling in the modelling technique for predicting set-up, current, or waves). Later, the discussion clearly focuseson the modelling technique, and the influence of coupling wave and currents from bothmodels. This issue is more technical, but also really interesting. Finally, the conclusioncomes back on the topic suggested in introduction. I suggest the authors slightlymodify introduction and conclusion to mention both type of results in introduction andconclusion.

Response: The first part of the above concern is also pointed out by reviewer-1. Accordingly, a common response has been prepared for the first part, and the same is given below:

The authors agree upon as pointed out by both the reviewers that this study focuses not only on the quantification of the impact of wave-current interaction, but also on: (i) impact of wave-current interaction on water level, (ii) impact of wave-current interaction on waves, and (iii) impact of wave-current interaction on currents. Accordingly, the last

paragraph of the Introduction section is modified as follows:

From literature review, it is evident that most of the studies carried out with storm surge models for the Indian coast used standalone models (Rao et al., 2012; Bhaskaran et al., 2014; Gayathri et al., 2015; Gayathri et al., 2016, Dhana Lakshmi et al., 2017). A comprehensive review on the coastal inundation research and an overview of the processes for the Indian coast was also reported by Gayathri et al. (2017). One can find very few studies reported using a coupled model (ADCIRC with SWAN) for the Indian seas (Bhaskaran et al., 2013; Murty et al., 2014, 2016; Poulose et al., 2017) for extreme weather events. These studies examined the performance of coupled models and role of improved wind forcing on waves and hydrodynamic conditions. The present study is a comprehensive exercise that aims to study the following interaction during the Hudhud event: (i) impact of wave-current interaction on water level, (ii) impact of wave-current interaction on waves, and (iii) impact of wave-current interaction on currents. This involves simulation of winds, tides, storm surges, currents and waves in the study domain during this extreme weather event using the coupled ADCIRC and SWAN models. Only the measured wave and water level data was available for the verification of model results (which happened to be very close to the cyclone track). Both these data sets were utilized in this study. Unfortunately, no measured current data was available for verification of the model-computed currents. The coupled model (ADCIRC+SWAN) has demonstrated its efficacy in predicting storm surge and water level elevation as compared to the standalone ADCIRC model. For example, considering the 2013 Phailin cyclone event (Murty et al., 2014), the difference in residual water level between standalone and coupled versions at Paradeep in Odisha coast were about 0.3m, and the coupled model performed relatively better than standalone model. In addition, for the 2011 Thane cyclone, good performance of coupled parallel ADCIRC-SWAN model was reported by Bhaskaran et al. (2013). The overall performance of waves and currents during Thane event validated against HF Radar observations and with satellite tracks of ENVISAT, JASON-1, JASON-2 and wave rider buoy observations very clearly show that coupled model performed reasonably well.

[Figure]

During extreme weather events like cyclones, the interaction between waves and currents is a highly non-linear process, and the transfer and exchange of energy between them is a very complex process. Along the nearshore regions, the non-linear interaction process is highly complex and to a larger extent, it is controlled by the local water depth and coastal geomorphological features. There can be instance wherein the computed results using a coupled model may be under-estimated considering the influence of currents. However, in this case the role of bottom characteristics and water level needs a separate detailed study. Also, including fine resolution bathymetry and cyclonic winds will further enhance the accuracy of the model.

References

Dhana Lakshmi, D., Murty, P.L.N., Bhaskaran, P.K., Sahoo, B., Srinivasa Kumar, T., Shenoi, S.S.C., Srikanth, A.S.: Performance of WRF-ARW winds on computed storm surge using hydrodynamic model for Phailin and Hudhud cyclones. Ocean Engineering, 131, 135-148, 2017.

Gayathri, R., Bhaskaran, P.K., Sen, D.: Numerical study on Storm Surge and associated Coastal Inundation for 2009 AILA Cyclone in the head Bay of Bengal. Int. Conf. on Water Resources, Coastal and Ocean Engineering (ICWRCOE 2015), Aquatic Procedia, 4, 404-411, 2015.

Gayathri, R., Murty, P.L.N., Bhaskaran, P.K., Srinivasa Kumar, T.: A numerical study of hypothetical storm surge and coastal inundation for AILA cyclone in the Bay of Bengal. Environmental Fluid Mechanics, 16(2), 429-452, 2016.

Gayathri, R., Bhaskaran, P.K., Jose, F.: Coastal inundation research: an overview of the process. Current Science, 112(2), 267-278, 2017.

Murty, P.L.N., Bhaskaran, P.K., Gayathri, R., Sahoo, B., Srinivasa Kumar, T., SubbaReddy, B.: Numerical study of coastal hydrodynamics using a coupled model for Hudhud cyclone in the Bay of Bengal. Estuarine, Coastal and Shelf Science, 183,

13-27, 2016.

Poulose, J., Rao, A.D., Bhaskaran, P.K.: Role of continental shelf on non-linear interaction of storm surges, tides and wind waves: An idealized study representing the west coast of India. Estuarine, Coastal and Shelf Science, http://dx.doi.org/10.1016/j.ecss.2017.06.007

The Conclusion section is modified in the revised manuscript as follows:

A coupled ADCIRC+SWAN modelling system has been used to simulate the changes that occurred in the ocean surface dynamics during the passage of Very Severe Cyclonic Storm Hudhud that made landfall near Visakhapatnam located on the East Coast of India. At the time of peak intensity, the Holland parametric model reproduced maximum wind speed of ïĊż54 m/s with a minimum central pressure drop of 950 hPa. The landfall of Hudhud event occurred during the spring high tide, and the tide gauge observation off Visakhapatnam recorded a maximum surge of 1.4 m, that matched reasonably well with the modelled surge (1.2 m). The two-way coupling with SWAN showed an increment of ïĊż0.25 m (20%) in the total water level elevation during this cyclone, which was contributed by waves to the total rise in water level. During the time of landfall near Visakhapatnam, the current speed increased from 0.5 m/s to 1.8 m/s for a short duration (ïĊż6 h) with the direction of flow towards south, and thereafter ïĊż 6 h the current speed reduced to ïĊż 0.1 m/s with a reversal in direction (towards north). The study signifies that an increase of ïĊż 0.2 m in significant wave height (Hs) was noted by including the effect of currents on the wave field. The inclusion of currents in the modelling system does have influence on the wave field, especially on wave length (in the present case, a change of about 2 s in wave period) and wave height. Increase in wave height was observed on the left side of the cyclone track, when waves and currents opposed each other (waves were propagating from southwest and currents flowing towards southwest). As wave-current interaction is a complex problem, and the expected changes in wave parameters are very small, further refinement is required in the two-way coupling of ADCIRC+SWAN (with fine resolution bathymetry and improved

cyclonic winds).

Comment – 3:

The modelling procedure for SWAN could be detailed a little bit more. The conditionsused here are really not usual conditions, and a commentary on how accurate theapproximations are in hurricane conditions would be welcomed.

Response: The authors appreciate the reviewer comments. The details of SWAN modelling are briefly given below:

SWAN (Simulating WAves Nearshore) is a third-generation wave model developed at the Delft University of Technology, Netherlands. It computes random, short-crested wind-generated waves in coastal regions and inland waters. The current version of SWAN is 40.85. The model is based on the wave action balance equation, with various source and sink mechanisms, that governs the redistribution of energy balance in the wave system. SWAN can be used on any scale relevant for wind generated surface gravity waves. However, the SWAN model is specifically designed for coastal applications that should actually not require such flexibility in scale. The input parameters that can be provided to SWAN includes bathymetry, current, water level, bottom friction and wind. The governing equation of SWAN is the wave action balance equation expressed in the form:

$$\partial N/\partial t+(\partial C_{g,x}\, N)/\partial x+(\partial C_{g,y}\, N)/\partial y+(\partial C_{g,\sigma}\, N)/\partial \sigma+(\partial C_{g,\theta}\, N)/\partial \theta=S/\sigma$$

where, N is the wave action density; ïĄş is the relative frequency; ïĄś is the wave direction; Cg is the propagation speed in (x,y,ïĄş,ïĄś) space; and S is the total of source/sink terms expressed as the wave energy density. In SWAN model the source terms are expressed in the form:

$$S=S_{in}+S_{ds,w}+S_{ds,b}+S_{nl4}+S_{nl3}$$

The terms in the R.H.S of the equation represents the wind input, white-capping, bottom friction, quadruplet wave-wave interactions and triad wave-wave interactions, respectively. The terms like bottom friction and triad wave-wave interaction can be neglected in deep water calculations. The model coupling is based on the work by Bunya et al. (2010) and Dietrich et al. (2011) conducted for the Gulf of Mexico region. The SWAN model employs an implicit sweeping method to update the wave field at each computational vertex, which allows SWAN to apply longer time steps than ADCIRC. Thus, the SWAN time step usually defines the coupling interval between SWAN and ADCIRC models (Dietrich, 2010; Dietrich et al., 2011a,b).

The wind field provided as input to SWAN model during Hudhud cyclone was generated using the Holland parametric model, which is specifically meant for simulating winds during cyclones.

Comment – 4:

Except for the wave buoy data (I would appreciate to see the location of the buoy ona map, by the way), the paper suffers a lack of data for validation. Could the author-saccess some other data, such as surface velocity from satellite, or water elevation fromPSMSL, for instance? It would help validating the numerical results.

Response: The authors appreciate the reviewer comments. The wave rider buoy location is plotted in Figure 1a. This was also pointed out by reviewer-1. It may kindly be noted that the water level elevation data off Visakhapatnam used in this study for validation of model results (Figure 4) is from PSMSL data only. However, no data is available for validation of currents, including satellite data during the passage of Hudhud cyclone at this location.

Comment – 5:

In section 3.4, I had difficulties to understand if the "SWAN alone" simulations wer-ereferring to SWAN with absolutely no current, or SWAN with input current from AD-CIRC,but no coupling. This clarification is obviously important for interpreting there-sults.

Response: The authors appreciate the reviewer comments. Two cases were run, viz, SWAN in standalone mode, and SWAN coupled with ADCIRC to assess the impact of currents on cyclone generated waves. SWAN alone simulations are referred to as simulation with no currents.

Comment – 6:

Minor points: - section 2.1, line 94. There is a misprint on the location of the wave rider buoy. Furthermore,I could not understand what the +20m -20m measurement range refers to. WaveHeight? It seems huge, and is probably not true regardless to the waves frequency.

Response: The authors appreciate the comments and thank the reviewer for pointing out this mistake. Also, as suggested by the other reviewer-1, the authors have modified the Section 2.1 with more detailed information of wave rider buoy as given below: The wave rider buoy location is corrected as: 17.63ïĆřN and 83.26ïĆřE. The measurement range +20m to -20m refers to the wave height with an accuracy of 3%. There were occasions when wave heights were in excess of 30 m, especially during very severe hurricanes.

The in situ data was recorded continuously at 1.28 Hz and the recording interval for every 30 min was processed as one record. At every 200 seconds a total number of 256 heave samples were collected and a Fast Fourier Transform (FFT) was applied to obtain a spectrum in the frequency range 0 to 0.58 Hz having a resolution of 0.005 Hz. Eight consecutive spectra covering 1600 seconds were averaged and used to compute the half-hourly wave spectrum. Significant wave height ($H_{m0}$) or $4\sqrt{m_0}$ was obtained from the wave spectrum. The nth order spectral moment ($m_n$) is given by: $m_n = \int_0^\infty âŰŠãĂŰf^n S(f)dfãĂŮ$, where $S(f)$ is the spectral energy density at frequency f. The period corresponding to the maximum spectral energy (i.e., spectral peak period ($T_p$) is estimated from the wave spectrum. The wave direction ($D_p$) and directional width corresponding to the spectral peak is estimated based on the circular

moments (Kuik et al.,1988).

Comment – 7:

- Section 3.1: Do we have estimation on how accurate Holland's numerical re-sultswere? Could the authors mention it with a sentence?

Response: The authors appreciate the reviewer comments. Hudhud cyclone reached the maximum intensity in the early morning of 12th October 2014 with a sustained wind speed of 180 km/h off Andhra coast. It crossed Visakhapatnam between 1200 and 1300 h IST on 12th October with the same wind speed (IMD Report, 2014). Figure 2 shows the passage of Hudhud cyclone, and the Holland model reproduced the max-imum wind speed of ï�ż186 km/h with a minimum central pressure drop of 950 hPa when it transformed into a Very Severe Cyclonic storm (Figure 2).

Comment – 8:

- For every figure, the captions are not detailed enough. Most of the time, it is unclear-what symbol corresponds to what line. The date and time used for various maps arenot mentioned.

Response: The authors appreciate the reviewer comments. Most of the suggested corrections are incorporated in the revised figures.

Fig. 1a. Bathymetry of the model domain chosen for wave-current interaction during Hudhud cyclone; cyclone track details are also shown; red dot represents wave rider buoy location. Fig. 1b. Fine resolution unstructured mesh generated for the domain to run the coupled ADCIRC+SWAN model; rectangular box represents the region where measured data are available for model validation (details of the box is shown in Fig. 1c). Fig. 1c. Fine-resolution mesh of the box shown in Fig. 1b; black circle is the landfall point of the Hudhud cyclone; cyclone track is also shown. Fig. 2. Typical winds (speed and direction) generated using Holland symmetrical model along the track of Hudhud cyclone (colour code represents wind speed in m/s; vectors represent wind direction).

Fig. 3. Spatial distribution of maximum surface elevation (m) due to (a) cyclonic winds, (b) cyclonic winds and tides and (c) cyclonic winds, tides and waves (colour code represents surface elevation in m). Fig. 4. Time series of surface elevation (m) representing measured surface elevation (red line), SE from ADCIRC alone (blue line) and SE from ADCIRC+SWAN (black line) at Visakhapatnam coast (17.63°N; 83.26°E) during 10-13 October 2014. Fig. 5. Spatial distribution of maximum surface currents (m/s) due to (a) winds, (b) winds and tides and (c) winds, tides and waves, during cyclone, (d) difference in current speeds from (b) and (c), illustrating change in current speeds due to wave-current interaction (colour code represents current speeds in m/s). Fig. 6. Time series of currents (m/s) representing current speeds and direction obtained from ADCIRC alone ('x' and blue rectangle) and coupled ADCIRC+SWAN ('+' and red rectangle) off Visakhapatnam coast (17.63°N; 83.26°E) during 10-13 October 2014. Fig. 7. Current speed and direction simulated along the track of Hudhud cyclone using the coupled ADCIRC+SWAN model (colour code represents current speed in m/s; vectors represent current direction). Fig. 8. Comparison of measured (black) and modelled (a) significant wave heights (Hs), (b) mean wave periods, (c) peak wave periods and (d) peak wave directions obtained from SWAN (red) and coupled ADCIRC+SWAN (blue) during Hudhud cyclone with measured data off Visakhapatnam (17.63°N; 83.26°E). Fig. 9. Significant wave heights (Hs) simulated along the track of Hudhud cyclone using the coupled ADCIRC+SWAN model (colour contours represent Hs in m). Fig.10. Spatial distribution of maximum significant wave heights (Hs) simulated along the track of Hudhud cyclone using (a) SWAN model (no wave-current interaction), (b) coupled ADCIRC+SWAN model (with wave-current interaction); colour code and contours represent Hs; (c) change in Hs from (a) and (b), illustrating change in wave energy due to wave-current interaction. Fig. 11. Spatial distribution of (a) mean wave period (Tm) and (b) peak wave period (Tp) simulated along the track of Hudhud cyclone using coupled ADCIRC+SWAN model (with wave-current interaction). Fig. 12. (a). Maximum radiation stress gradient values calculated from SWAN and (b) spatial distribution of mean wave direction (Dir) simulated along the track of Hudhud cyclone using the coupled ADCIRC+SWAN model (with wave-current interaction); colour code and contours represent wave direction.

Please note that figure is included in the supplement file.

Please also note the supplement to this comment: https://www.nat-hazards-earth-syst-sci-discuss.net/nhess-2017-11/nhess-2017-11-AC2-supplement.pdf

---

## Author Response (AR1)

**Comment – 1:**

The manuscript has a subject that is appropriate to the NHESS publication and should be of interest to readers of the journal. The paper is reasonably well written, logicallyorganised, structured and illustrated. The authors present an interesting set of simulations assess the mutual affects of waves and currents during the passage of the Very Severe Cyclonic Storm Hudhud. The principal results are: (1) Waves contributed 0.25 m to the total water level during the event, which agrees with measurements atVisakhapatnam; (2) Current speed increased from 0.5 to 1.8 m/s for a short time during the event; (3) the two-way coupling increased the current magnitude by 0.25 m/salong the track; (4) The use of wave-ocean coupling increases Hs in 2 m compared to wave model only; (5) waves decrease due to currents when they travel normal to he coast after crossing the shelf area (right side of track) and increased on the leftside of track when currents oppose wave direction. Cyclonic systems currently poseone of the most challenging and important meteo-oceanographic phenomena for theearth science community and the study is a valuable contribution. However I havefundamental objections that I believe should be addressed before final publication.

**Response:**

The authors thank the reviewer for thoroughly reviewing our manuscript, appreciating the work and providing valuable suggestions to improve the manuscript. The fundamental objections raised by the reviewer have been addressed in the following paragraphs.

**Comment – 2:**

My main concern is related to the overall content and discussions. In the end of introduction, the authors state that "The present study primarily aims at quantifying theimpact of wave-current interaction on waves during the Hudhud cyclone". The analysishowever seems to be equally focused on the effects of waves on currents and waterlevel. I presume they do so because the only data source available is of a Wave Riderbuoy. No current data is available. However,

there is no clear discussion on whetherthe inclusion of current improves wave simulation at the buoy location. The authorsjust mention the differences in model results show the plots of comparison of modeland measurement and let the readers draw their own conclusion. The coupling systemincrease wave height (0.2 m) at the wave height peak moment. But wave height isdecreased before this moment and model actually agrees better with data without theinclusion of currents. Wave period is also slightly better represented in the simulationwithout currents. If the main goal of the paper is "quantifying" the effect of wave-current and/or deterioration of simulations compared to measurements. At least an attempt should bemade. It is an interesting opportunity to address some limitations of these models and currents are actually beneficial to wave modelling (and vice-versa).

**Response:**

The authors appreciate the reviewer comments and agree that this study focus not only on the quantification pertaining to the impact of wave-current interaction, but also on: (i) impact of wave-current interaction on water level, (ii) impact of wave-current interaction on waves and (iii) impact of wavecurrent interaction on currents. Accordingly, the last paragraph in the Introduction section has been modified, and relevant references were added as follows:

[revised manuscript text omitted]

**Comment – 3:**

The exact location of the wave rider buoy must be plotted, possibly on the map of figure 1a, so that the reader can know where the validation of the wave model was performed for.

**Response:**

The authors appreciate the reviewer comments. Accordingly as suggested, the buoy location is marked in Figure 1a.

**Comment – 4:**

Very little detail is given about wave measurements. Although section 2 is entitled"Data and Methodology" it is basically about the modelling configuration. What is thesampling time of wave information, how is it obtained (spectral method, record length)?

This information together with the aforementioned plot of the buoy location may be of interest to readers.

**Response:**

The authors appreciate the reviewer comments. The wave data used in this study was obtained from the National Institute of Ocean Technology, Chennai. As suggested by the reviewer, the details of wave measurements and data analysis are now added in the revised manuscript as follows:

Wave data was obtained from the directional wave rider buoy deployed off Visakhapatnam (17.63°N; 83.26°E) at 15 m water depth. The measurement range is -20 m to 20 m, with an accuracy of 3%. The in situ data was recorded continuously at 1.28 Hz and the recording interval for every 30 min was processed as one record. At every 200 seconds, a total number of 256 heave samples were collected and a Fast Fourier Transform (FFT) was applied to obtain a spectrum in the frequency range 0 to 0.58 Hz having a resolution of 0.005 Hz. Eight consecutive spectra covering 1600 seconds were averaged and used to compute the half-hourly wave spectrum. Significant wave height ( $H_{m0}$ ) or  $4\sqrt{m_0}$  was obtained from the wave spectrum. The nth order spectral moment ( $m_n$ ) is given by:  $m_n = \int_0^{\infty} f^n S(f) df$ , where S(f) is the spectral energy density at frequency f. The period corresponding to the maximum spectral energy (i.e., spectral peak period ( $T_p$ ) is estimated from the wave spectrum. The spectral peak is estimated based on the circular moments (Kuik et al., 1988).

**Reference:**

Kuik, A.J., Vledder, G., Holthuijsen, L.H., A method for the routine analysis of pitch and roll buoy wave data, Journal of Physical Oceanography 18, 1020–1034, 1988.

**Comment – 5:**

Why do the authors decide for the set of physics of growth and dissipation fromCavaleri and Malanotte-Rizzoli (1981) and Komen et al. (1984)? These old parameterisationsand especially the Komen et al. dissipation form are proven to not be adequatein non-standard conditions, as in opposing currents, for example (see Ardhuin et al JPO(2012) and Rapizo et al (JGR 2017)). The dissipation form in Westhuysen et al (2007)shows better performance than the Komen et al. term in adverse currents (Rapizoet al, 2017). The newly implemented in SWAN and recently released 'ST6' physics(Rogers et al, 2012) performs best in conditions of effective currents Rapizo et al (JGR2017), which is the subject of investigation here. If the old physics are used instead, ajustification must be given.

**Response:**

The authors appreciate the reviewer comments. The authors have conducted this study in 2015 using the unstructured version of SWAN (version 40.85) implementing an analog to the four-direction Gauss-Seidel iteration technique with unconditional stability (Zijlema, 2010). However, Rapizo et al (2017) reported the good performance of SWAN in tidal current regime (ebb and flood flows) very recently (2017) only. It may kindly be noted that, the co-author of this work, Bhaskaran and his team has carried out a few studies (Bhaskaran et al., 2014; Gayathri et al., 2015; Gayathri et al., 2016, Dhana Lakshmi et al., 2017; Bhaskaran et al., 2013; Murty et al., 2014, 2016; Poulose et al., 2017) using the same formulation of Komen et al. (1984) for cyclones that occurred in the Indian Ocean region, and found that SWAN with this scheme performed well for extreme weather events also. Keeping this in view, in the present study the authors have gone ahead with using the same formulation of Komen et al to study the wave-current interaction during the Hudhud event. However, the authors appreciate the reviewer comments and shall use the scheme of Roger et al (2012) in SWAN and study the wave-current interaction in tidal as well in cyclonic conditions as a separate study in future.

**Comment – 6:**

I find it hard to analyse the differences in current speed shown in Fig. 5 (especiallyfor figure (b)). Although it is interesting to see the pattern produced by the cyclonelandfall, all figures show similar patterns. I suggest here to plot (b) and (c) as currentspeed differences (similar to Fig. 10 bottom panels).

**Response:**

The authors appreciate the reviewer comments. As suggested one more Figure (5d) is added to show the difference in current speed similar to Fig. 10 in the revised manuscript.

**Comment – 7:**

Other minor points: Line 22: "Studies show that waves contribute to local currents, water level and mixing." By mentioning "Studies" I feel at least one reference is neededhere.

**Response:**

The authors appreciate the reviewer comments. As suggested the following three references are added to this statement in the revised manuscript:

Kudryavtsev et al., 1999; Davies and Lawrence, 1995; McWilliams et al., 2004. These studies show that waves contribute to local currents, water level and mixing.

**Comment – 8:**

Line 24-26: "Several studies have been carried out relating to individual processes, but not many on interaction between the processes. Therefore, we need to take intoaccount different processes that impact a specific process." Very confusing, manyrepetitions of word "process". Rephrase.

**Response:**

The sentence is rephrased as follows in the revised manuscript: 'Several studies have been carried out relating to individual processes, but not on interactions between them'.

**Comment – 9:**

Line 34: "effected" => "affected"

**Response:**

The correction made accordingly in the revised manuscript.

**Comment – 10:**

Line 37: "The wave processes that impact the coastal environment are:" There aremany other wave-related processes that impact the coastal environment other thanthe ones listed (wave set-up, wave-current interactions and breaking-induced mixing). The first phrase should be rephrased to something like: "Some of the relevant waveprocesses that impact the coastal environment are as follows:"

**Response:**

The sentence is modified as follows in the revised manuscript: "Some of the wave processes that impact the coastal environment are as follows:"

**Comment – 11:**

Line 55: SWAN stands for Simulating Waves Nearshore, not "in Nearshore".

**Response:**

The correction is made accordingly in the revised manuscript.

**Comment – 12:**

Line 93: Buoy coordinates are wrong. (same for legend in Fig. 4, 6 and 8)

**Response:**

The authors appreciate the reviewer comments. As suggested, the corrections are made (17.63°N; 83.26°E) in the revised manuscript.

**Anonymous Referee #2**

Received and published: 6 June 2017

**Comment – 1:**

The paper at hand is interesting, and addresses an important topic, namely the influenceof wave-current interactions on water set up, current magnitude, and water waveevolution in very strong conditions. The case studied here is the Hudhud cyclone.Obviously, the topic is relevant for publication in Natural Hazards and Earth SystemSciences. Furthermore, the paper is clear and well structured. It is relatively wellwritten, and pleasant to read. For these reasons, I believe it should be published inNHESS. However, I have some concerns, which, if addressed, could help improve thepaper.

**Response:**

The authors thank the reviewer for thoroughly reviewing our manuscript, appreciating the work and providing the positive recommendations. Based on the valuable suggestions, the authors have improved the manuscript. The response to comments and concerns are addressed below.

**Comment – 2:**

The introduction states that "The present study primarily aims at quantifying the impactof wave-current interaction on waves during the Hudhud cyclone". But the paperpresents more results than this (effect of wave-current coupling in the modelling technique for predicting set-up, current, or waves). Later, the discussion clearly focuses on the modelling technique, and the influence of coupling wave and currents from bothmodels. This issue is more technical, but also really interesting. Finally, the conclusioncomes back on the topic suggested in introduction. I suggest the authors slightlymodify introduction and conclusion to mention both type of results in introduction and conclusion.

**Response:**

The first part of the above concern is also pointed out by reviewer-1. Accordingly, a common response has been prepared for the first part, and the same is given below:

The authors agree upon as pointed out by both the reviewers that this study focuses not only on the quantification of the impact of wave-current interaction, but also on: (
[revised manuscript text omitted]

**Comment – 3:**

The modelling procedure for SWAN could be detailed a little bit more. The conditions here are really not usual conditions, and a commentary on how accurate the approximations are in hurricane conditions would be welcomed.

**Response:**

The authors appreciate the reviewer comments. The details of SWAN modelling are briefly given below:

SWAN (Simulating WAves Nearshore) is a third-generation wave model developed at the Delft University of Technology, Netherlands. It computes random, short-crested wind-generated waves in coastal regions and inland waters. The current version of SWAN is 40.85. The model is based on the wave action balance equation, with various source and sink mechanisms, that governs the redistribution of energy balance in the wave system. SWAN can be used on any scale relevant for wind generated surface gravity waves. However, the SWAN model is specifically designed for coastal applications that should actually not require such flexibility in scale. The input parameters provided to SWAN includes bathymetry, current, water level, bottom friction and wind. The governing equation of SWAN is the wave action balance equation is expressed in the following form:

$$\frac{\partial N}{\partial t} + \frac{\partial C_{g,x}N}{\partial x} + \frac{\partial C_{g,y}N}{\partial y} + \frac{\partial C_{g,\sigma}N}{\partial \sigma} + \frac{\partial C_{g,\theta}N}{\partial \theta} = \frac{S}{\sigma}$$

where, N is the wave action density;  $\sigma$  is the relative frequency;  $\theta$  is the wave direction; Cg is the propagation speed in  $(x,y,\sigma,\theta)$  space; and S is the total of source/sink terms expressed as the wave energy density. In SWAN model the source terms are expressed in the following form:

$$S = S_{in} + S_{ds,w} + S_{ds,b} + S_{nl4} + S_{nl3}$$

The terms in the R.H.S of the equation represent wind input, white-capping, bottom friction, quadruplet wave-wave interactions and triad wave-wave interactions, respectively. The terms like bottom friction and triad wave-wave interaction can be neglected in deep water calculations. The model coupling is based on the work of Bunya et al. (2010) and Dietrich et al. (2011) conducted for the Gulf of Mexico region. The SWAN model employs an implicit sweeping update the wave field each computational method to at vertex. which allows SWAN to apply longer time steps than ADCIRC. Thus, the SWAN time step usually defines the coupling interval between SWAN and ADCIRC models (Dietrich, 2010; Dietrich et al., 2011a,b).

The wind field provided as input to SWAN model during Hudhud cyclone was generated using the Holland parametric model, which is specifically meant for simulating winds during cyclones.

**Comment – 4:**

Except for the wave buoy data (I would appreciate to see the location of the buoy ona map, by the way), the paper suffers a lack of data for validation. Could the authorsaccess some other data, such as surface velocity from satellite, or water elevation fromPSMSL, for instance? It would help validating the numerical results.

**Response:**

The authors appreciate the reviewer comments. The wave rider buoy location is plotted in Figure 1a. This was also pointed out by reviewer-1. It may kindly be noted that the water level elevation data off Visakhapatnam used in this study for validation of model results (Figure 4) is from PSMSL data only. However, no data is available for validation of currents, including satellite data during the passage of Hudhud cyclone at this location.

---

## Referee Report (RR1)

The authors addressed most of my suggested corrections and concerns. However my main concern was not addressed properly in my view. I feel compelled to quote here a passage of my first review: " … However, there is no clear discussion on whether the inclusion of current improves wave simulation at the buoy location. The authors just mention the differences in model results show the plots of comparison of model and measurement and let the readers draw their own conclusion. The coupling system increase wave height (0.2 m) at the wave height peak moment. But wave height is decreased before this moment and model actually agrees better with data without the inclusion of currents. Wave period is also slightly better represented in the simulation without currents. If the main goal of the paper is "quantifying" the effect of wave-current interactions on waves as stated in the introduction, this must be discussed also in terms of improvement and/or deterioration of simulations compared to measurements. At least an attempt should be made. It is an interesting opportunity to address some limitations of these models and if currents are actually beneficial to wave modelling (and vice-versa)."

Other than this, all points were properly corrected or modified according to the corrections/suggestions.

---

## Referee Report (RR2)

I see the change made by the authors in the new manuscript was the inclusion of lines 252-258. By doing so the authors attempted to address my concern regarding the inclusion of a proper discussion on the advantages of the use of currents in the wave model. Although the discussion added in the current version improved the manuscript and compare both simulations (with and without currents), my point from the last two revisions was not addressed whatsoever. There are still no discussion on whether the inclusion of current improves or deteriorates wave simulation at the buoy location. It, once again, simply mention that currents reduce wave height and period and that the overall effect is marginal. The final conclusion on the relevance and benefits of currents to the simulations are left to be drawn by the readers.

My two simple questions to the authors are: Do the inclusion of currents improve wave simulation? Are the current-induced reduction in wave height and period beneficial to wave simulation in hurricane Hudson? This is the same point I raised in my previous two revisions, which I feel compelled to raise once again, since it was not properly addressed. I suggest the use of statistical metrics of bias and errors to answer the above questions in a quantitative sense.

---

## Author Response (AR2)

**Reply to Referee 2**

The authors addressed most of my suggested corrections and concerns. However my main concern was
not addressed properly in my view. I feel compelled to quote here a passage of my first review:
" … However, there is no clear discussion on whether the inclusion of current improves wave
simulation at the buoy location. The authors just mention the differences in model results show the plots
of comparison of model and measurement and let the readers draw their own conclusion. The coupling
system increase wave height (0.2 m) at the wave height peak moment. But wave height is decreased
before this moment and model actually agrees better with data without the inclusion of currents. Wave
period is also slightly better represented in the simulation without currents. If the main goal of the
paper is "quantifying" the effect of wave-current interactions on waves as stated in the introduction, this must
be discussed also in terms of improvement and/or deterioration of simulations compared to
measurements. At least an attempt should be made. It is an interesting opportunity to address some
limitations of these models and if currents are actually beneficial to wave modelling (and vice-versa)."
Other than this, all points were properly corrected or modified according to the corrections/suggestions.

**We apologize for not replying to the above comment of the reviewer satisfactorily. Now, we have
discussed the above concern referring to Figs. 7 and 8 as follows:**

**Referring to Fig.8, we find that more or less the measured significant wave heights match with the
modelled wave heights (with and without currents near the buoy location, off Vishakapatnam).
When current was introduced, wave heights reduced approximately by 0.2m and mean wave
periods reduced by 2s. It may be noted that during this time, the waves and currents were nearly
in the same direction (Figs. 7 and 8d). Subsequently, when current speed increased to 0.5 m/s (Fig.
6) during 1300h to 2000h (12th October 2014) with the wave and currents directions opposite to
each other, we observe an increase in wave height of approximately 0.3m. Hence, there is an
influence of currents on waves though it is marginal.**

**Accordingly, the above response is now included in the manuscript (page No 9; section 3.4), and
the text is revised as follows:**

[revised manuscript text omitted]

---

## Author Response (AR3)

**Reply to Referee**

I see the change made by the authors in the new manuscript was the inclusion of lines 252-258. By doing so the authors attempted to address my concern regarding the inclusion of a proper discussion on the advantages of the use of currents in the wave model. Although the discussion added in the current version improved the manuscript and compare both simulations (with and without currents), my point from the last two revisions was not addressed whatsoever. There are still no discussion on whether the inclusion of current improves or deteriorates wave simulation at the buoy location. It, once again, simply mention that currents reduce wave height and period and that the overall effect is marginal. The final conclusion on the relevance and benefits of currents to the simulations are left to be drawn by the readers.

My two simple questions to the authors are: Do the inclusion of currents improve wave simulation? Are the current-induced reduction in wave height and period beneficial to wave simulation in hurricane Hudson? This is the same point I raised in my previous two revisions, which I feel compelled to raise once again, since it was not properly addressed. I suggest the use of statistical metrics of bias and errors to answer the above questions in a quantitative sense.

**Response to Comment – 1:**

**The authors appreciate the referee for the constructive comments. The results obtained for the Hudhud cyclone show that inclusion of currents does not improve the wave simulation. It is found that inclusion of currents deteriorated the wave simulation at the buoy location when waves and currents were nearly in the same direction, whereas when waves and currents were in the opposite direction, the inclusion of currents enhanced the wave simulation. Overall, it is seen that the influence of currents on the wave system is marginal. This observation is also supported by a recent study of Liu et al. (2016), and that is now included in the revised manuscript.**

[revised manuscript text omitted]